# Reducing Shape-Radiance Ambiguity in Radiance Fields with a Closed-Form Color Estimation Method

**Qihang Fang**[1,2,*]     **Yafei Song**[3,*]     **Keqiang Li**[1,2]     **Liefeng Bo**[3]

[1]State Key Laboratory of Multimodal Artificial Intelligence Systems, Institute of Automation, Chinese Academy of Sciences
[2]School of Artificial Intelligence, University of Chinese Academy of Sciences
[3]Alibaba Group
[1,2]{fangqihang2020,likeqiang2020}@ia.ac.cn
[3]{huaizhang.syf,liefeng.bo}@alibaba-inc.com

## Abstract

A neural radiance field (NeRF) enables the synthesis of cutting-edge realistic novel view images of a 3D scene. It includes density and color fields to model the shape and radiance of a scene, respectively. Supervised by the photometric loss in an end-to-end training manner, NeRF inherently suffers from the shape-radiance ambiguity problem, *i.e.,* it can perfectly fit training views but does not guarantee decoupling the two fields correctly. To deal with this issue, existing works have incorporated prior knowledge to provide an independent supervision signal for the density field, including total variation loss, sparsity loss, distortion loss, *etc*. These losses are based on general assumptions about the density field, *e.g.*, it should be smooth, sparse, or compact, which are not adaptive to a specific scene. In this paper, we propose a more adaptive method to reduce the shape-radiance ambiguity. The key is a rendering method that is only based on the density field. Specifically, we first estimate the color field based on the density field and posed images in a closed form. Then NeRF's rendering process can proceed. We address the problems in estimating the color field, including occlusion and non-uniformly distributed views. Afterwards, it is applied to regularize NeRF's density field. As our regularization is guided by photometric loss, it is more adaptive compared to existing ones. Experimental results show that our method improves the density field of NeRF both qualitatively and quantitatively. Our code is available at https://github.com/qihangGH/Closed-form-color-field.

## 1   Introduction

A neural radiance field (NeRF) [20, 4, 23, 8, 32] is a cutting-edge technique in computer graphics and computer vision that enables the synthesis of realistic novel view images of a 3D scene from a collection of posed 2D images. It represents a 3D scene with a density and a color field, which describe the scene's shape and radiance, respectively. Despite its impressive novel-view synthesis capability, the NeRF suffers from an inherent shape-radiance ambiguity problem [33]. To be more specific, a family of degradation solutions exist, which can perfectly recover training views but produce incorrect shape. As a consequence, poor novel views would be rendered.

The shape-radiance ambiguity exists because a NeRF-based method usually renders an image using the Max volume rendering algorithm [17]. It integrates the density and color fields closely. As shown in Fig. 1(a), a pixel is rendered as the sum of products of density and color values. In general, such a rendering process is supervised by the photometric loss in an end-to-end training manner, where the

---

*Both authors contributed equally to this work.

37th Conference on Neural Information Processing Systems (NeurIPS 2023).

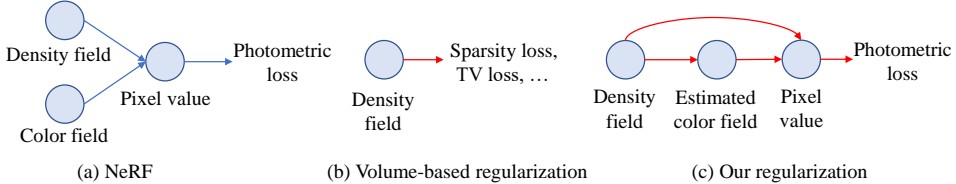

Figure 1: Graphical representations of the NeRF and its regularizations.

density and color fields do not receive independent supervision signals. As a result, NeRF can either modify its density or color field to reduce the loss. Unfortunately, performing the latter may well fit the training views, but neglect the error in shape.

To reduce the ambiguity, previous work focuses on different aspects. A group of methods utilizes additional assumptions, *e.g.*, sparse or dense depth priors [5, 25, 22, 29], to bypass the influence of the color field and directly supervise the density field. A group of other methods improves NeRF's representation, *e.g.*, by modeling the surface normal [30], to inherently reduce the ambiguity. Besides, a group of methods incorporates prior belief to the original NeRF's representation for regularization. Typical ones are volume-based loss, such as sparsity [10, 7] and total variation (TV) loss [26, 7], and ray-based loss, such as background entropy loss [15, 27], ray-entropy loss [13], and distortion loss [1]. These losses directly supervise the density field with reasonable prior knowledge, so that the shape-radiance ambiguity can be alleviated. We typically focus on the third group of methods and their effects on explicit NeRF models, which are well-known for their training and rendering efficiency. Despite the efficacy of learning a better density field, most existing regularizations are not aware of the geometry of a scene. They are based on some general assumptions, *e.g.*, the space tends to be sparse, the density should distribute smoothly in the space, and it should distribute compactly along a ray. Thus, such regularizations may be less adaptive or too aggressive.

In this paper, we propose a more adaptive method to regularize the density field, which elegantly breaks the entanglement of the shape and radiance in NeRFs. Specifically, we devise a rendering method that is *only based on the density field* as shown in Fig. 1(c). The key is to first estimate the color field given the density field and posed training images by using predefined rules. Then, the NeRF's rendering process can proceed as usual. Notably, the color field is not learnable here, and so the rendering results solely depend on the density field. In this case, the photometric loss can well supervise the scene shape. Calculating the color field from a density field, however, faces several hurdles, such as occlusion and non-uniformly distributed views. We tackle them by weighting the observed colors with transmittance and using a residual color estimation scheme, respectively. Overall, our regularization is more adaptive to specific scene geometry compared with existing volume and ray-based ones, as it is guided by the photometric loss. The results of two typical explicit NeRFs, including Plenoxels [7] and Direct Voxel Grid Optimization (DVGO) [27, 28], on the NeRF Synthetic [20], LLFF [19], and DTU datasets [12] demonstrate the superiority of our method. Overall, our contributions are summarized as follows:

1. We propose a closed-form color field estimation method given the density field and posed images. We deal with problems that include occlusion and non-uniformly distributed views. Experimental results show that the estimated color field has acceptable rendering quality.

2. We use the estimated color field for regularization, and reduce shape-radiance ambiguity accordingly. Our method is aware of scene shape and so more adaptive. Experimental results show that our loss improves the shape both qualitatively and quantitatively.

3. We implement our regularization method with CUDA kernels. Thus, the computational cost for training a scene is still acceptable.

## 2  Related Work

**Novel-view synthesis**. The NeRF [20] has revolutionized the novel-view synthesis technology [24] with its impressive rendering quality. It uses fully-connected neural networks to map a 3D location and viewing direction to a density and view-dependent color. Then a novel view is synthesized by using the MAX volume rendering algorithm [17]. Besides the implicit neural network representation,

explicit NeRFs [14, 7, 27, 28, 21, 2] are proposed to accelerate both training and rendering speed. The density and color field are represented by voxel grids, which store the density values or the color features. In this way, a lightweight multilayer perceptron (MLP) should suffice for predicting the view-dependent colors. Furthermore, researchers have paid great effort to optimize the rendering quality in challenging scenarios [1, 16, 18], training speed [14, 7, 27, 28, 21, 31], and inference speed [11, 10, 3, 23, 8, 32]. Despite these improvements, the NeRF's representation of a scene inherently suffers from the shape-radiance ambiguity problem [33]. A complex color field can well fit the training images while leaving an error in geometry. Although Zhang *et al.* [33] analyzes that the NeRF relieves the problem by using an MLP with limited capacity and low-frequency positional encoding components, the shape-radiance ambiguity problem needs to be solved more systematically.

**Regularization methods for the density field**. The regularization methods can be divided into volume-based and ray-based ones. The most simple volume-based regularization is the sparsity loss [10, 32, 7]. It is based on the fact that most of the space in a scene is empty. Another effective regularization is the TV loss [26, 7], which assumes that the density values of adjacent voxels should not change drastically. It helps increase the smoothness of the density field. Ray-based regularization methods, on the other hand, impose prior beliefs on how the density along a ray should distribute. The background entropy loss [15, 27] forces the final transmittance of a ray to be either 1 or 0, which means that the ray can only belong to the background or the foreground. This can effectively sharpen the contour of an object. The ray-entropy loss [13] is an improved version of the background entropy loss. It argues that the density values along a ray should have a low entropy. This is achieved by minimizing the Shannon entropy of the normalized ray density. Such a regularization helps produce better geometry than the vanilla NeRF. The distortion loss [1] shares a similar idea with the ray-entropy loss. It encourages the density along a ray to distribute as compactly as possible. This is achieved by optimizing the non-empty intervals to be narrow and near to each other. Experimental results show that the distortion loss effectively removes the floaters in the free space.

Overall, we notice that the above volume and ray-based methods are all based on general assumptions. They are directly applied to the density field, and so may be less adaptive. In this paper, we propose a more adaptive method to regularize the density field, which is guided by the photometric loss.

## 3 Preliminaries

A radiance field is composed of a density field $\mathcal{F}^\sigma$ and a color field $\mathcal{F}^c$. The density $\mathcal{F}^\sigma_{\mathbf{v}} \in \mathbb{R}$ of a point $\mathbf{v} \in \mathbb{R}^3$ indicates the differential probability of a ray hitting a particle at $\mathbf{v}$, and $\mathcal{F}^c_{\mathbf{v},\mathbf{d}} \in \mathbb{R}^3$ represents the color emitted by $\mathbf{v}$ along the direction $\mathbf{d}$. The NeRF uses volume rendering to render the color $\hat{C}(\mathbf{r})$ of a ray $\mathbf{r}(t) = \mathbf{o} + t\mathbf{d}$ that starts at a camera's original point $\mathbf{o}$ following the Max [17] rendering equation

$$\hat{C}(\mathbf{r}) = \int_{t_n}^{t_f} T(t)\mathcal{F}^\sigma_{\mathbf{r}(t)}\mathcal{F}^c_{\mathbf{r}(t),\mathbf{d}}\, dt, \tag{1}$$

where $t_n$ and $t_f$ are near and far bounds for the integral, and the $T(t)$ is the transmittance defined as

$$T(t) = \exp\left(\int_{t_n}^{t} -\mathcal{F}^\sigma_{\mathbf{r}(s)} ds\right). \tag{2}$$

To deal with the integral, the NeRF assumes that the density and color are piece-wise constant along the ray $\mathbf{r}$. Thus, the integrals above become summations.

Given a collection of images and their corresponding camera poses. A photometric loss $\mathcal{L}_p$ is applied to minimize the difference between a rendered pixel color $\hat{C}(\mathbf{r})$ and its ground-truth pixel value $C(\mathbf{r})$ as follows,

$$\mathcal{L}_p = \frac{1}{|\mathcal{R}|}\sum_{\mathbf{r}\in\mathcal{R}}\left|\left|C(\mathbf{r}) - \hat{C}(\mathbf{r})\right|\right|_2^2, \tag{3}$$

where $\mathcal{R}$ is a mini-batch of rays.

In such a formulation, let $\mathcal{F}^{wc}_{\mathbf{r}(t),\mathbf{d}} = T(t)\mathcal{F}^\sigma_{\mathbf{r}(t)}\mathcal{F}^c_{\mathbf{r}(t),\mathbf{d}}$ denotes the transmittance-density-weighted color field, the photometric loss in effect optimizes $\mathcal{F}^{wc}_{\mathbf{r}(t),\mathbf{d}}$ to fit training views. An optimal $\mathcal{F}^{wc}_{\mathbf{r}(t),\mathbf{d}}$ learned on training views, however, does not guarantee a correct density field, since it can be achieved by infinitely possible combinations of density and color fields. This raises the shape-radiance ambiguity problem.

# 4 Method

The method is divided into two parts. First, we elaborate on the color field estimation method given a density field and posed images. Then we use the estimated color field for regularization.

## 4.1 Closed-Form Color Field Estimation

The key problem in this subsection is to estimate a color field $\hat{\mathcal{F}}^c_{\mathbf{v},\mathbf{d}}$ as accurately as possible given a density field $\mathcal{F}^\sigma_{\mathbf{v}}$ and posed observation images.

**Spherical harmonics representation for the color field**. We use spherical harmonics (SH) to represent the color of each point. The SH includes a series of orthogonal and complete basis functions defined on the surface of the unit sphere $\mathbf{S}$ [9]. Following [7], a linear combination of SH basis functions is used to approximate the color of a point $\mathbf{v}$ observed from a direction $\mathbf{d}$ as

$$\mathcal{F}^c_{\mathbf{v},\mathbf{d}} = \sum_{\ell=0}^{L} \sum_{m=-\ell}^{\ell} h_\ell^m Y_\ell^m(\mathbf{d}), \tag{4}$$

where $L$ is the degree of the SH basis, $Y_\ell^m(\mathbf{d})$ is the SH function with a degree $\ell$, order $m$, and $h_\ell^m$ is its coefficient. Note that $h_\ell^m$ is a function of $\mathbf{v}$. We omit its notation for brevity. Under this formulation, the color estimation problem is equivalent to estimating the SH coefficients.

Similar to Fourier series, the coefficient of a specific frequency band can be obtained by integrating the product of the color and the corresponding SH function as follows,

$$\begin{aligned}
h_\ell^m &= \int_{\mathbf{d}\in\mathbf{S}} \mathcal{F}^c_{\mathbf{v},\mathbf{d}} Y_\ell^m(\mathbf{d}) d\mathbf{d} \\
&= \int_{\mathbf{d}\in\mathbf{S}} p_{\mathbf{d}}(\mathbf{d}) \frac{\mathcal{F}^c_{\mathbf{v},\mathbf{d}} Y_\ell^m(\mathbf{d})}{p_{\mathbf{d}}(\mathbf{d})} d\mathbf{d} \\
&\approx \frac{1}{K} \sum_{k=1}^{K} \frac{\mathcal{F}^c_{\mathbf{v},\mathbf{d}_k} Y_\ell^m(\mathbf{d}_k)}{p_{\mathbf{d}}(\mathbf{d}_k)}, \quad \mathbf{d}_k \sim p_{\mathbf{d}}(\mathbf{d}),
\end{aligned} \tag{5}$$

where $p_{\mathbf{d}}(\mathbf{d})$ is the probability density function of directions, and the integral is approximated by Monte Carlo sampling. In our case, we assume that the direction should be uniformly distributed and $p_{\mathbf{d}}(\mathbf{d}) \equiv \frac{1}{4\pi}$. However, we cannot randomly select the direction since its color could not be observed. Actually, we only know the samples $\mathbf{d}_k$ from posed images.

**Transmittance-weighted color for occlusion handling**. The posed images provide a set of $\{\mathbf{d}_k, \mathcal{F}^c_{\mathbf{v},\mathbf{d}_k}\}$ for Eq. (5). We use a colored rectangle as an example scene as shown in Fig. 2, where $\mathbf{v}$ is a point on its surface. Specifically, there are four cameras with centers $\mathbf{o}_k, k = 1, 2, 3, 4$, and the direction $\mathbf{d}_k = \frac{\mathbf{o}_k-\mathbf{v}}{\|\mathbf{o}_k-\mathbf{v}\|}$ is a normalized vector that connects $\mathbf{o}_k$ and $\mathbf{v}$. Assume that the projection matrix of the $k^{th}$ camera is $\mathbf{P}_k$, the $\mathcal{F}^c_{\mathbf{v},\mathbf{d}_k}$ is the observation color at $\mathbf{P}_k\mathbf{v}$ of the image plane.

Ideally, if there is no occlusion, the radiance emitted by $\mathbf{v}$ can be directly captured by a camera ($\mathbf{o}_2$ and $\mathbf{o}_3$). Then, the radiance is the same as the corresponding observation color ($\mathcal{F}^c_{\mathbf{v},\mathbf{d}_2}$ and $\mathcal{F}^c_{\mathbf{v},\mathbf{d}_3}$). However, it is possible that the point $\mathbf{v}$ is occluded ($\mathbf{o}_0$ and $\mathbf{o}_1$), in which case the observation color ($\mathcal{F}^c_{\mathbf{v},\mathbf{d}_0}$ and $\mathcal{F}^c_{\mathbf{v},\mathbf{d}_1}$) is not the radiance emitted by $\mathbf{v}$, but the radiance of a surface point that is closest to the camera center. These observations should be treated differently when estimating the SH coefficients. We resort to the transmittance $T_{\mathbf{v},k}$ between $\mathbf{v}$ and $\mathbf{o}_k$, *i.e.*,

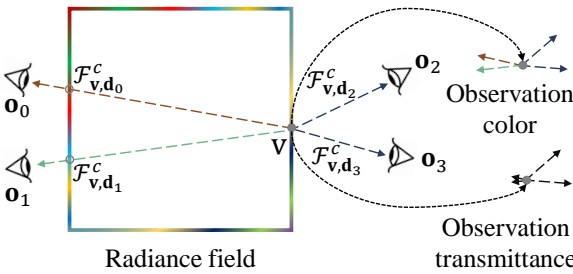

Figure 2: Observation colors and transmittance.

$$T_{\mathbf{v},k} = \exp\left(\int_0^{\|\mathbf{o}_k-\mathbf{v}\|} -\mathcal{F}^\sigma_{\mathbf{r}(s;\mathbf{v},\mathbf{d}_k)} \, ds\right), \tag{6}$$

where $\mathbf{r}(s;\mathbf{v},\mathbf{d}_k) = \mathbf{v} + s\mathbf{d}_k$. As $T_{\mathbf{v},k}$ exponentially accumulates the density along a ray, an occluded ray has a very low transmittance. Thus, we use it as a weight factor and modify Eq. (5) as follows,

$$\hat{h}_\ell^m = \frac{1}{\sum_{k=1}^{K} T_{\mathbf{v},k}} \sum_{k=1}^{K} \frac{T_{\mathbf{v},k} \mathcal{F}_{\mathbf{v},\mathbf{d}_k}^c Y_\ell^m(\mathbf{d}_k)}{p_{\mathbf{d}}(\mathbf{d}_k)}. \tag{7}$$

**Residual estimation for non-uniformly distributed views**. The estimation given by Eq. (7) is strongly affected by non-uniformly distributed sample directions. To calculate the coefficient of a frequency band, the integral in Eq. (5) relies on the orthogonality of the SH basis that

$$\int_{\mathbf{d}\in\mathbf{S}} Y_\ell^m(\mathbf{d}) Y_i^j(\mathbf{d}) d\mathbf{d} = \begin{cases} 1, & i = \ell \text{ and } j = m, \\ 0, & \text{otherwise.} \end{cases} \tag{8}$$

When we discretize the integral with samples that are non-uniformly distributed on the sphere $\mathbf{S}$, Eq. (8) may be violated. For simplicity, we use the product of two trigonometric functions $\cos 0x \cdot \cos x$ as an example. As shown in Fig. 3, the positive and negative values in uniform samples better compensate each other to approach a zero-value summation, while the non-uniform case generates a large bias.

To reduce the estimation bias, we propose a residual estimation scheme. The basic idea is to subtract the terms that integrate to zero in the integral to reduce their influences. Specifically, we further modify Eq. (7) as follows,

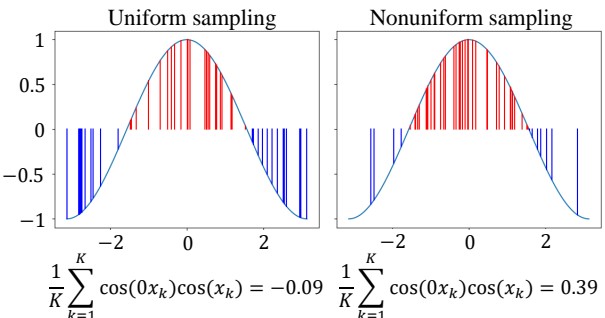

$$\frac{1}{K}\sum_{k=1}^{K}\cos(0x_k)\cos(x_k) = -0.09 \qquad \frac{1}{K}\sum_{k=1}^{K}\cos(0x_k)\cos(x_k) = 0.39$$

Figure 3: Non-uniform samples generate a larger bias.

$$\hat{h}_\ell^m = \frac{1}{\sum_{k=1}^{K} T_{\mathbf{v},k}} \sum_{k=1}^{K} \frac{T_{\mathbf{v},k} \cdot {}^m_\ell \tilde{\mathbf{c}}_k \cdot Y_\ell^m(\mathbf{d}_k)}{p_{\mathbf{d}}(\mathbf{d}_k)}, \tag{9}$$

with ${}^m_\ell \tilde{\mathbf{c}}_k$ defined as

$${}^m_\ell \tilde{\mathbf{c}}_k = \mathcal{F}_{\mathbf{v},\mathbf{d}_k}^c - \sum_{i=0}^{\ell} \sum_{j=-i}^{\tilde{m}} \hat{h}_i^j \cdot Y_i^j(\mathbf{d}_k). \tag{10}$$

where $\tilde{m} = m$ if $i = \ell$, else $\tilde{m} = i$. This scheme can effectively reduce estimation biases. Please refer to the Appendix A for more explanations.

## 4.2 Regularization with the Estimated Color Field

We use the estimated color field $\hat{\mathcal{F}}_{\mathbf{v},\mathbf{d}}^c$ to reduce the shape-radiance ambiguity. Specifically, we can render a closed-form pixel $C_{cf}(\mathbf{r})$ with $\hat{\mathcal{F}}_{\mathbf{v},\mathbf{d}}^c$ and $\mathcal{F}_{\mathbf{v}}^\sigma$ following the volume rendering equation as per Eq. (1). Then, a closed-form photometric loss (CF loss for short) can be calculated as follows,

$$\mathcal{L}_{cf} = \frac{1}{|\mathcal{R}_{cf}|} \sum_{\mathbf{r}\in\mathcal{R}_{cf}} ||C(\mathbf{r}) - C_{cf}(\mathbf{r})||_2^2, \tag{11}$$

where $\mathcal{R}_{cf}$ is a mini-batch of rays. With a weight factor $\lambda$, the regularization term $\lambda\mathcal{L}_{cf}$ is added to the training loss for reducing the shape-radiance ambiguity.

# 5 Experiments

## 5.1 Experimental Settings

**Datasets**. To evaluate our method, we conduct experiments on DTU [12], NeRF synthetic [20], and LLFF datasets [19]. Each DTU scene has 49 or 64 images. We evenly select 5 or 6 images

from them as testing ones. Besides, we downsample the $1600 \times 1200$ images to be $800 \times 600$ ones. For NeRF synthetic and LLFF datasets, we follow the conventional train-test split used in previous studies. By default, we use a black background for DTU and NeRF synthetic datasets, as it makes the shape-radiance ambiguity more severe on some scenes.

**Baselines**. We choose two explicit NeRFs, including Plenoxels [7] and DVGO [27, 28], as baselines. Among them, the Plenoxels has been trained with the sparsity loss and TV loss, and DVGO has been trained with the background entropy loss.

**Metrics**. Besides the commonly applied peak signal-to-noise ratio (PSNR) metric of images, we calculate the PSNR of depth for the NeRF Synthetic dataset to evaluate the geometry, since it provides the ground truth depth. Depth values are normalized into the range $[0, 1]$ before calculating the PSNR. Additionally, we adopt the metric named inverse mean residual color (IMRC) [6] to evaluate the geometry of a scene. It is defined as the transmittance-density-weighted residual color in dB as follows

$$\text{IMRC} = -10 \log_{10} \frac{\sum_{\mathbf{v} \in \mathcal{V}} \sum_k T_{\mathbf{v},k} \left(1 - \exp\left(-\mathcal{F}^\sigma_{\mathbf{v}} \cdot \delta_{\mathbf{v}}\right)\right) \left(\frac{L}{L}\tilde{\mathbf{c}}_{\mathbf{v},k}\right)^2}{\sum_{\mathbf{v} \in \mathcal{V}} \sum_k T_{\mathbf{v},k} \left(1 - \exp\left(-\mathcal{F}^\sigma_{\mathbf{v}} \cdot \delta_{\mathbf{v}}\right)\right)}, \tag{12}$$

where $T_{\mathbf{v},k}$ is the transmittance defined in Eq. (6), $\frac{L}{L}\tilde{\mathbf{c}}_{\mathbf{v},k}$ is the final residual color defined in Eq. (10), and $\delta_{\mathbf{v}}$ is the half-width of a voxel as the step size. To get deeper insights into Eq. (12), it is a weighted mean squared error of the radiance emitted from all directions $\mathbf{d}_k$ of all points $\mathbf{v}$. The error or residual color $\frac{L}{L}\tilde{\mathbf{c}}_{\mathbf{v},k}$ of a point $\mathbf{v}$ at a direction $\mathbf{d}_k$ is the difference between the ground truth observation color and recovered radiance based on the geometry. The better geometry should better recover the color field. Thus, the higher the IMRC, the lower the residual color, the better.

**Implementation details**. The codes of Plenoxels and DVGO are borrowed from the official release. A uniform probability distribution $p_{\mathbf{d}}(\mathbf{d})$ of directions is applied. In other words, $p_{\mathbf{d}}(\mathbf{d}) \equiv \frac{1}{4\pi}$ is a constant value. Please refer to Appendix B.1.1 for more discussions about the probability density function of directions. We implement the CF loss with CUDA kernels. For explicit NeRFs, the 3D space is divided into voxel grids. We need to estimate the color field for the voxels. Specifically, the block number is set as the number of voxels. Each block has 128 threads and each thread handles a ray that connects the voxel and a camera. Following [7], we set the SH degree to 2, *i.e.*, there are $3 * (2 + 1)^2 = 27$ SH coefficients per voxel. For Plenoxels, the number of rays $|\mathcal{R}_{cf}|$ is set to 25 for all datasets. The weight factor $\lambda$ is set to 10, 0.1, and 0.5 for the DTU, NeRF synthetic, and LLFF dataset, respectively. For DVGO, the number of rays $|\mathcal{R}_{cf}|$ is set to 10 for all datasets. The weight factor $\lambda$ is set to 1 and 0.1 for the coarse and fine stage of the DTU dataset, 2 and 0.1 for the coarse and fine stage of the NeRF synthetic dataset, and 0.1 for the LLFF dataset. All the experiments are run on a single NVIDIA Tesla A100 GPU.

## 5.2  Results of the Closed-Form Color Field Estimation Given Trained Density Fields

In this subsection, we verify the effectiveness of the closed-form color estimation. Specifically, we train Plenoxels and DVGO with their original implementations without adding the CF loss. Then, the density fields of these trained models are used to estimate color fields without any training process.

The occlusion handling and residual estimation play significant roles. We conduct an ablation study here. The results based on Plenoxels' density fields are reported in Table 1. In Fig. 4, we additionally showcase an example based on a DVGO's density field. Specifically, when we directly estimate SH coefficients according to Eq. (7) without using the residual estimation scheme, the rendered images look bad as shown by the first two sub-figures. This is because estimating the SH coefficient of one frequency band is affected by other frequency components if we cannot uniformly sample directions from the 2D sphere. Since the images of each DTU scene are approximately taken from a half sphere, the views are non-uniformly distributed. As a result, the estimation of SH coefficients suffers from biases. On the other hand, when we estimate

Table 1: Ablation study on DTU scenes based on Plenoxels' density.

| Occlusion handling | Residual estimation | PSNR |
|:---:|:---:|:---:|
|  |  | 14.61 |
| ✓ |  | 13.57 |
|  | ✓ | 25.99 |
| ✓ | ✓ | **26.49** |

without occlusion handling, *i.e.*, without using transmittance to weight an observation, the estimation result is blurred as shown by the third sub-figure. This is because in the estimation, $\mathcal{F}^c_{\mathbf{v},\mathbf{d}_k}$ of different $\mathbf{d}_k$ is not guaranteed to be the radiance emitted from the same point $\mathbf{v}$, but possibly that of a point

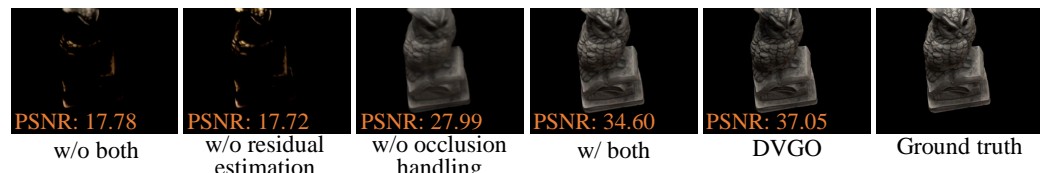

| w/o both | w/o residual estimation | w/o occlusion handling | w/ both | DVGO | Ground truth |
|----------|-------------------------|------------------------|---------|------|--------------|
| PSNR: 17.78 | PSNR: 17.72 | PSNR: 27.99 | PSNR: 34.60 | PSNR: 37.05 | |

Figure 4: Ablation study on DTU scan 122 based on DVGO's density.

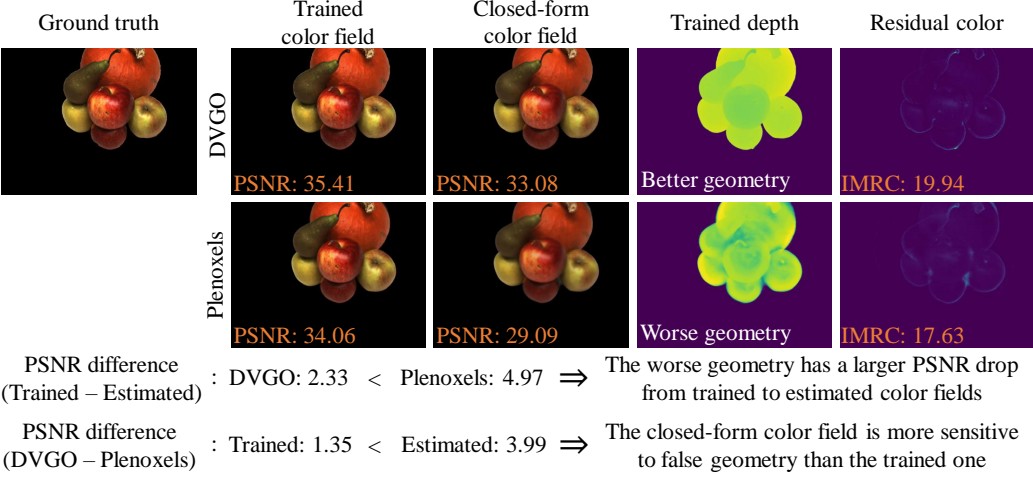

Figure 5: Color field estimation results of DTU scan 63.

nearer to a camera. When both occlusion handling and residual estimation are applied, as shown in the fourth sub-figure, the rendering result is visually acceptable, and quantitatively, has a much higher PSNR that approaches the trained one. To summarize, both the table and figure show that the residual estimation improves the PSNR considerably. With only occlusion handling, the PSNR decreases, but when it is combined with the residual color estimation, the PSNR is further improved.

The estimation results are not only affected by the algorithm, but also by the quality of the density field. Specifically, the estimation is based on an assumption that, if a point is not occluded, its radiance along a direction is directly determined by the corresponding observation color in the image. This assumption is true for the ideal case, where the density is sharp enough and the color reduces to the surface light field [33]. By using this assumption, the closed-form color field becomes sensitive to false geometry. If the geometry or the density field is not correct or sharp enough, which breaks this assumption, the color estimation results will be bad. In Fig. 5, we show the estimation results based on the density field of DVGO and Plenoxels, respectively. We also visualize the depth map of the scene and render the residual color. The Plenoxels suffers a sharper PSNR drop from the trained image (34.06) to estimated one (29.09). This is because it has an inferior density field as qualitatively shown by the depth maps. The errors in the density field affect the estimation process and result in poor rendering quality. In contrast, a better density field given by the DVGO permits a higher PSNR (33.08) of the estimated color field. Note that the closed-form color field is more sensitive to the quality of density field than the trained one, as it produces a larger PSNR difference ($3.99 > 1.35$) between two methods. By using the closed-form photometric loss, the rendering errors caused by an inferior density field can be back-propagated to improve the density field.

## 5.3 Results of Regularization and Comparison

In this subsection, we analyze the regularization results. Specifically, we apply our regularization to Plenoxels and DVGO and train from scratch. Based on DVGO, we additionally add a fine-tuned distortion loss for comparison. Table 2 reports the PSNR, IMRC, and PSNR of depth.

For Plenoxels, the CF loss improves all metrics on all datasets. It effectively improves the geometry of a scene. As shown in Fig. 6, the floaters in the free space (marked by red circles) are penalized,

Table 2: Comparison of PNSR ↑/IMRC ↑/(PSNR of depth↑) on the three datasets (**Best**).

| Method | DTU | NeRF synthetic | LLFF |
|---|---|---|---|
| Plenoxels [7] | 31.86 / 15.88 | 29.83 / 16.55 / 22.70 | 26.30 / 19.15 |
| Plenoxels [7] + CF loss | **32.08** / **16.66** | **29.84** / **16.59** / **23.08** | **26.40** / **20.02** |
| DVGO [27, 28] | 32.15 / 18.19 | 31.58 / 17.25 / 25.46 | 26.23 / 18.87 |
| DVGO [27, 28] + Distortion loss [1] | 32.20 / 18.47 | 31.50 / 17.93 / 25.68 | 26.33 / 20.76 |
| DVGO [27, 28] + CF loss | 32.23 / 18.51 | **32.24** / 17.61 / 26.36 | 26.26 / 19.66 |
| DVGO [27, 28] + Distortion loss [1] + CF loss | **32.26** / **18.80** | 32.23 / **18.90** / **26.68** | **26.34** / **20.92** |

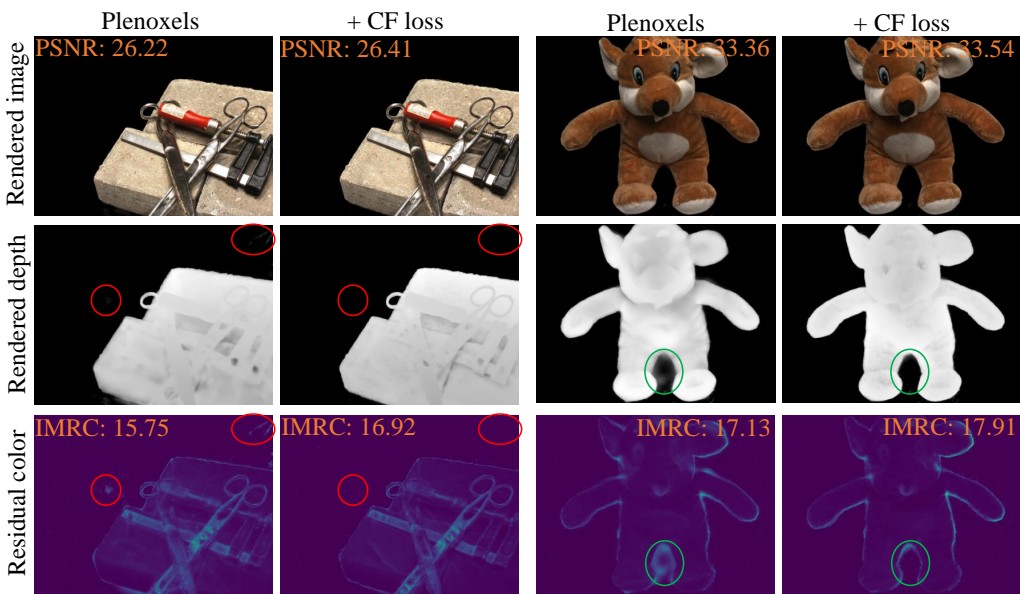

Figure 6: Our loss improves the geometry of a scene by making the density field sharper and removing the floaters in the free space. The two scenes are DTU scan 37 and 105 from left to right.

and the density field becomes sharper (marked by green circles). This is reasonable because the observation colors of the points in wrong geometry, *e.g.*, floaters or a thick surface, tend to be very high frequency. The SH of degree 2 fails to fit such a high-frequency color variation, and the estimated color field may produce poor rendering results, as represented by a high CF loss. During back-propagation, the CF loss can guide to a better density field.

For DVGO, the best results for all metrics on all datasets are obtained when the CF loss is added or both CF loss and distortion loss are added. These two losses complement each other's advantages, but the CF loss is more sensitive to specific geometric errors. We visualize the rendered images, depth, and residual color of several scenes for better comparison. As shown in Fig. 7, the DVGO fails to recover two details of a house as marked by blue and red rectangles. The distortion loss relieves the problems, but the windows in the red rectangle are still distorted. We try to further increase the weight factor of the distortion loss but this does not help. In contrast, our loss significantly improves the quality of the density field as shown by the depth maps and residual colors. Two more examples are shown in Fig. 8. It is noteworthy that the scene with a mic is trained with abundant views (100 views) uniformly distributed in the upper hemisphere. Under a black background, however, it severely suffers from the shape-radiance ambiguity problem. A part of the wire is confused with the background, and lots of floaters exist around the wire. The distortion loss makes no difference as it is not aware of the geometry of the scene. The CF loss successfully distinguishes the wire and background, producing a better density field. The right scene in Fig. 8 is another example. The DVGO, with or without distortion loss, mistakenly reconstructs the upper surface of the bottom rock. This mistake is avoided by our method. Overall, the CF loss effectively improves the geometry of a scene qualitatively and quantitatively.

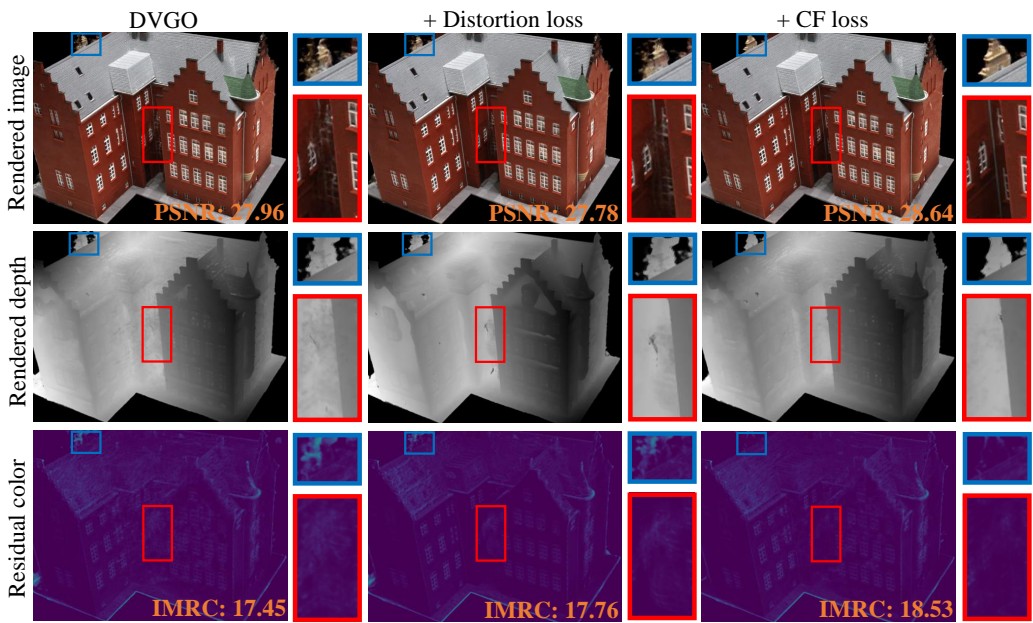

Figure 7: Comparison of distortion loss and CF loss on DTU scan 24.

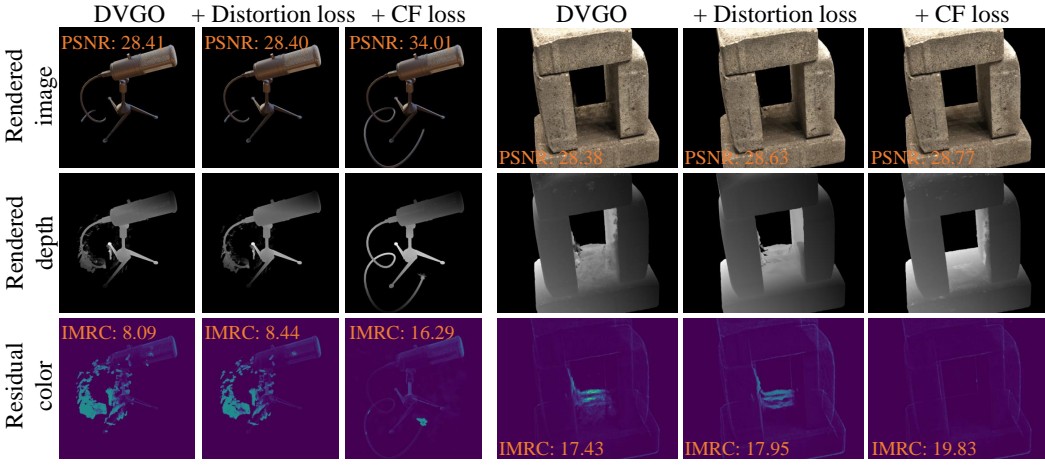

Figure 8: Comparison of distortion loss and CF loss on the mic from the NeRF synthetic dataset and DTU scan 40.

### 5.4 Computational Cost

In this subsection, we analyze the computational cost. Specifically, in each training step, a batch of $B$ rays are used. Then, only the colors of the voxels intersected with these rays need to be estimated. Suppose that each ray involves the order of $N$ voxels, there are $M = BN$ voxels. To estimate the color of a voxel, the density values along the ray from the voxel to $K$ source cameras need to be calculated. Then the total number of voxels involved is of order $KMN = KBN^2$. In our implementation, each CUDA block handles one of the $M$ voxels, and each thread handles one of the $K$ cameras. As the threads are executed in parallel, in theory, this

Table 3: Training time (minutes) with different batch sizes ($B$).

| $B$ | DTU | NeRF synthetic | LLFF |
|---|---|---|---|
| 0 | 17.7 | 16.0 | 24.2 |
| 10 | 32.7 | 20.5 | 29.4 |
| 25 | 43.0 | 25.5 | 34.6 |

reduces the computation overhead from the order of $KBN^2$ to $N$. In practice, we observe that sometimes the computational overhead is affected by the batch size. This is due to the hardware limit of the maximum CUDA threads. When an excessive number of threads are used, they are not guaranteed to be executed in parallel. In Table 3, we report the computational cost of training Plenoxels by using different batch sizes for regularization. The number of source cameras $K$ used are $44$ or $58$ for the DTU, $100$ for NeRF synthetic, and from $17$ to $54$ for LLFF dataset. The DTU dataset requires longer training time because it is trained for 12 epochs, while the NeRF synthetic dataset is trained for 9 epochs. For 9 epochs, the DTU dataset costs about $24$ and $32$ minutes for $10$ and $25$ rays, respectively. Compare these with those of NeRF synthetic dataset, we can observe that the large number of $K = 100$ cameras does not obviously increase the computational burden.

## 6   Conclusion

The shape-radiance ambiguity is a significant problem of NeRFs. Even with a simple background and abundant training images, wrong geometry can be learned and poor novel-views can be rendered. In this paper, we propose a new regularization method to break the entanglement of density and color field. The key is a closed-form color estimation method, which can recover the color field of a scene given a density field and a set of posed training images. We overcome the difficulties in estimating the color fields by handling the occlusion and non-uniformly distributed views. The image quality rendered by our closed-form color field approaches the trained one. Then, we use the photometric loss derived from the estimated color field to provide an independent supervision signal for the density field. Both the PSNR and IMRC of two explicit NeRFs, including Plenoxels and DVGO, are improved. Experimental results show the capability of the CF loss to correct geometric errors compared with existing volume and ray-based losses.

Ground truth    Trained color field    Closed-form color field

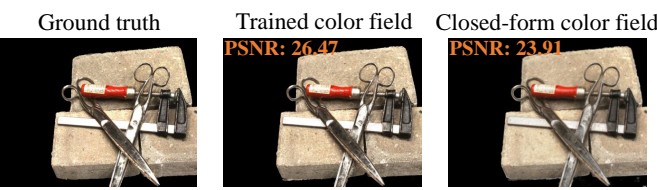

Figure 9: Color estimation results of a highly reflective object given the trained DVGO density.

**Limitations and future work**. The closed-form color field has difficulties to perfectly recover highly reflective objects. As shown in Fig. 9, the surface of the scissors is less reflective compared to the ground truth. Although increasing the SH degree can relieve this problem to some extent as shown in Appendix B.1.2, it needs to be solved more systematically. For example, the probability density function $p_\mathbf{d}(\mathbf{d})$ of directions can be better estimated, and a rigorous proof of the bias reduction strategy needs to be further studied. Besides, a well-estimated color field has the potential to get rid of the parameterized color field in a NeRF, and then only the density field needs to be trained and stored.

## Acknowledgements

This work was supported in part by the National Key Research and Development Program of China under Grant 2021YFB3301504, in part by the National Natural Science Foundation of China under Grant U19B2029, Grant U1909204, and Grant 92267103, in part by the Guangdong Basic and Applied Basic Research Foundation under Grant 2021B1515140034, and in part by the CAS STS Dongguan Joint Project under Grant 20211600200022.

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

# Appendix

## A    More Explanations for the Residual Color Estimation Scheme

We first shed light on why the residual color estimation scheme helps reduce the bias caused by nonuniform sampling. Then we conduct numerical experiments to support our claim. For simplicity, the explanation is based on using Fourier series to approximate one-dimensional signals. The basic idea is the same and can be easily extended to the case that uses spherical harmonics (SH) to approximate the signals defined on a spherical surface.

Suppose that the target signal $f(x)$ defined on $[-\pi, \pi]$ is "good" enough, or satisfies the Dirichlet Conditions, so it could be expanded into Fourier series, *i.e.*,

$$f(x) = A_0 + \sum_{n=1}^{\infty} \left[ a_n \cos(nx) + b_n \sin(nx) \right]. \tag{13}$$

Given a set of samples $\{(x_t, f(x_t)) | t = 1, ..., T\}$, we aim to estimate the Fourier coefficients up to a typical degree $k_{max}$. The calculation of the coefficient $a_k$ is well-known as

$$
\begin{aligned}
a_k &= \frac{1}{\pi} \int_{-\pi}^{\pi} f(x) \cos(kx) dx \\
&= \frac{1}{\pi} \int_{-\pi}^{\pi} \left\{ A_0 \cos(kx) + \sum_{n=1}^{\infty} [a_n \cos(nx) \cos(kx) + b_n \sin(nx) \cos(kx)] \right\} dx.
\end{aligned}
\tag{14}
$$

Using the Monte Carlo method, the above integral can be approximated by $T$ samples $x_1, ..., x_T$ as

$$\hat{a}_k = \frac{1}{\pi} \frac{2\pi}{T} \sum_{t=1}^{T} \left\{ A_0 \cos(kx_t) + \sum_{n=1}^{\infty} [a_n \cos(nx_t) \cos(kx_t) + b_n \sin(nx_t) \cos(kx_t)] \right\}. \tag{15}$$

Because of the orthogonal completeness of the trigonometric function set, if uniform sampling is performed and the number of samples is large enough, all other terms in Eq. (15) will approach $0$ except for $a_k \cos(kx_t) \cos(kx_t)$. Moreover, according to the Monte Carlo method, we have

$$\int_{-\pi}^{\pi} \cos(kx) \cos(kx) dx \approx \frac{2\pi}{T} \sum_{t=1}^{T} \cos(kx_t) \cos(kx_t). \tag{16}$$

And then,

$$\hat{a}_k \approx \frac{1}{\pi} \frac{2\pi}{T} \sum_{t=1}^{T} a_k \cos(kx_t) \cos(kx_t) = a_k \frac{1}{\pi} \frac{2\pi}{T} \sum_{t=1}^{T} \cos(kx_t) \cos(kx_t) \approx a_k \frac{1}{\pi} \pi = a_k. \tag{17}$$

In the nonuniform sampling case, however, non-negligible estimation biases will be generated. As an example, we visualize the bias generated by the direct current component $\frac{1}{T} \sum_{t=1}^{T} A_0 \cos(x_t)$ in Fig, 10. In the uniform sampling case, $\frac{1}{T} \sum_{t=1}^{T} A_0 \cos(x_t)$ is better approaching $0$, as the positive (red) and negative (blue) function values are well counteracted. In contrast, the nonuniform sampling case induces a larger bias as controlled by the magnitude of $A_0$. We show such a bias in a numerical experiment later.

To reduce the bias caused by $\sum_{t=1}^{T} A_0 \cos(kx_t)$, we find it effective to subtract this component in the estimation, *i.e.*,

$$
\begin{aligned}
a_k &= \frac{1}{\pi} \int_{-\pi}^{\pi} f(x) \cos(kx) dx \\
&= \frac{1}{\pi} \int_{-\pi}^{\pi} (f(x) - A_0) \cos(kx) dx \\
&= \frac{1}{\pi} \int_{-\pi}^{\pi} \left\{ \sum_{n=1}^{\infty} [a_n \cos(nx) \cos(kx) + b_n \sin(nx) \cos(kx)] \right\} dx.
\end{aligned}
\tag{18}
$$

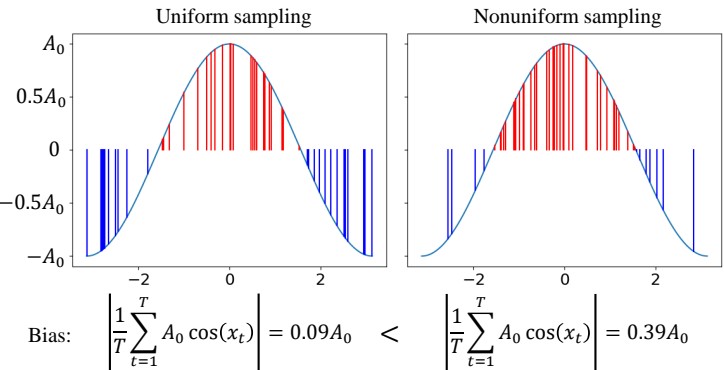

Figure 10: Nonuniform sampling generates a larger estimation bias than that of uniform sampling.

This time the Monte Carlo estimator of $a_k$ becomes

$$\hat{a}'_k = \frac{2}{T} \sum_{t=1}^{T} \left\{ \sum_{n=1}^{\infty} [a_n \cos(nx_t) \cos(kx_t) + b_n \sin(nx_t) \cos(kx_t)] \right\}. \tag{19}$$

Compared with $\hat{a}_k$ defined in Eq. (15), $\hat{a}'_k$ completely eliminates the bias caused by $\sum_{t=1}^{T} A_0 \cos(kx_t)$. Thus, the Monte Carlo estimator

$$\frac{2}{T} \sum_{t=1}^{T} (f(x_t) - A_0) \cos(kx_t) \tag{20}$$

as derived from $\frac{1}{\pi} \int_{-\pi}^{\pi} (f(x) - A_0) \cos(kx) dx$ suffers less bias from nonuniform sampling than

$$\frac{2}{T} \sum_{t=1}^{T} f(x_t) \cos(kx_t) \tag{21}$$

as derived from $\frac{1}{\pi} \int_{-\pi}^{\pi} f(x) \cos(kx) dx$. Because $A_0$ is actually unknown, we first estimate this direct current component using the Monte Carlo method as

$$\hat{A}_0 = \frac{1}{T} \sum_{t=1}^{T} f(x_t). \tag{22}$$

Then, we substitute the unknown $A_0$ with $\hat{A}_0$ in Eq. (20).

Note that, the bias of other components that are integrated to 0 could be analysed and eliminated similarly. We show in the experiments later that such a treatment effectively reduces the estimation bias.

As a summary, when estimating the coefficient $a_k$ of a frequency component, we should subtract all other frequency components that are integrated to 0 to reduce the estimation bias as much as possible. Notably, the coefficients for frequency components $k+1, k+2, ..., k_{max}$ are unknown if we perform the estimation in turn from 1 to $k$. One feasible solution is to perform the estimation iteratively. Concretely, in the first round, we only subtract the frequency components 1 to $k-1$ when estimating $a_k$. Then in the second round, we can subtract all other components. The iteration is ended until the difference of the estimated coefficients in the adjacent round becomes small enough. In the main paper, only one round of estimation is performed considering the computational overheads. **Notice that because the SH functions also has orthogonal completeness, the above scheme, *i.e.*, estimating the coefficients in turn and eliminating the influences of other frequency components, also makes a difference for estimating SH coefficients as we have done in the main paper**.

We perform numerical experiments about Fourier series to support our claim. We compare three estimators, including the original Monte Carlo estimator, Monte Carlo estimator using our bias

reducing strategy, and the estimator based on the least square method. Specifically, the least square solution is

$$
\begin{pmatrix}
1 & \cos(x_1) & \sin(x_1) & \cdots & \cos(k_{max}x_1) & \sin(k_{max}x_1) \\
1 & \cos(x_2) & \sin(x_2) & \cdots & \cos(k_{max}x_2) & \sin(k_{max}x_2) \\
\vdots & \vdots & \vdots & \ddots & \vdots & \vdots \\
1 & \cos(x_T) & \sin(x_T) & \cdots & \cos(k_{max}x_T) & \sin(k_{max}x_T)
\end{pmatrix}^{\dagger}
\begin{pmatrix}
f(x_1) \\
f(x_2) \\
\vdots \\
f(x_T)
\end{pmatrix}, \tag{23}
$$

where $\dagger$ denotes pseudo-inverse. We make sure that the number of samples $T$ is larger than the number of coefficients to be estimated.

For the target signals, we randomly choose from trigonometric and polynomial functions, and determine their coefficients randomly. As a result, three random target functions are chosen as follows,

$$
\begin{aligned}
f_1(x) &= 2 + 0.03x^2 + 2\sin(x) + \cos(3x), \\
f_2(x) &= 10 - 0.02x + 0.01x^2 + \cos(x) - \sin(2x), \\
f_3(x) &= 5 + 0.05x^2 - 0.001x^3 - \sin(x) + 2\cos(2x).
\end{aligned} \tag{24}
$$

To perform nonuniform sampling, we sample $x_t$ from a normal distribution $\mathcal{N}(0, \sigma^2)$. To compare the three estimators quantitatively, we uniformly sample 1000 points from $[-2\sigma, 2\sigma]$. Then, we calculate the root mean squared error (RMSE) of this 1000 points. Because of the randomness of the estimation process, we repeat the estimation for $100,000$ times and report the mean RMSE (MRMSE) as the final metric, *i.e.*,

$$
\text{MRMSE} = \frac{1}{100000} \sum_{i=1}^{100000} \sqrt{\frac{1}{1000} \sum_{j=1}^{1000} \left( f(x_{ij}) - \hat{f}_i(x_{ij}) \right)^2}, \tag{25}
$$

where $x_{ij}$ is the $j^{th}$ sample in the $i^{th}$ estimation process. $f(\cdot)$ is the ground-truth signal and $\hat{f}_i(\cdot)$ is the estimated signal in the $i^{th}$ estimation process. The lower the MRMSE, the better.

We set $k_{max} = 3$, $\sigma = 10$, and the number of samples for estimation $T = 10, 20, ..., 100$. The MRMSE results for $f_1(x)$, $f_2(x)$, and $f_3(x)$ are shown in Fig. 11. It is clear that with our bias reducing strategy, the estimation is consistently better than original Monte Carlo estimation. Furthermore, in the case of sparse sampling, the least square method suffers a sharp performance degradation. When only 10 samples are available, it is even worse than the original Monte Carlo estimation. When more samples are given, our method and the least square method become comparable.

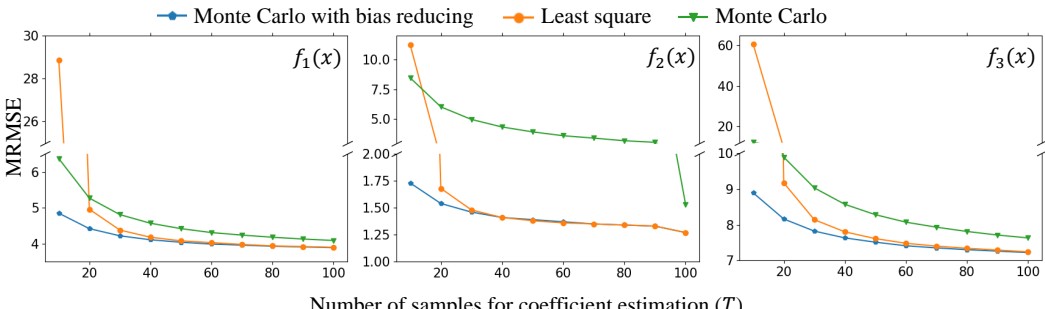

Figure 11: The MRMSE↓ results for $f_1(x)$, $f_2(x)$, and $f_3(x)$.

In addition, we add different direct current values to $f_1(x)$ to increase its $A_0$, and fix $T = 100$. The MRMSE results are listed in Table 4. It shows that the estimation bias of Monte Carlo estimation increases with a larger $A_0$, while our method and the least square method are invariant about such additions.

Finally, the computational cost of our method, least square method, and original Monte Carlo method for one estimation with 100 samples are $1.196 \times 10^{-4}$ s, $1.052 \times 10^{-4}$ s, and $0.978 \times 10^{-4}$ s respectively. The experiments are performed on an Intel i7-9700 CPU. Overall, we can conclude that our method effectively reduces estimation biases with acceptable computational overhead.

Table 4: The MRMSE↓ results when adding different direct current values to $f_1(x)$.

| Addition | 0 | 1 | 5 | 10 | 50 | 100 |
|---|---|---|---|---|---|---|
| Monte Carlo with bias reducing | 3.89 | 3.89 | 3.89 | 3.89 | 3.89 | 3.89 |
| Least square | 3.90 | 3.90 | 3.90 | 3.90 | 3.90 | 3.90 |
| Monte Carlo | 4.09 | 4.17 | 4.60 | 5.31 | 13.53 | 25.13 |

# B  More Experimental Results

## B.1  More Closed-Form Color Estimation Results

### B.1.1  Probability Density Functions for Directions

A common choice for modeling the distribution of 3D directions is to use the von Mises-Fisher (vMF) distribution defined as follows,

$$v(\mathbf{d}; \mu, c) = \frac{c}{2\pi(e^c - e^{-c})e^{c\mu^\top \mathbf{d}}}, \tag{26}$$

where $\mu$ is the normalized mean direction, and $c$ is the concentration parameter. Because the vMF distribution only deals with the unimodal case, we resort to the mixture of von Mises-Fisher distribution defined as

$$p(\mathbf{d}; \mathbf{d}_1, ..., \mathbf{d}_K, c) = \frac{1}{K} \sum_{k=1}^{K} v(\mathbf{d}_k; \mu, c), \tag{27}$$

where $\mathbf{d}_1, ..., \mathbf{d}_K$ are known viewing directions as the modes of the distribution. It is noteworthy that when $c$ approaches zero, the probability density function becomes a uniform distribution. Thus, the uniform distribution used in the main paper is a special case of the mixture of vMF distribution. The concentration parameter $c$ is a hyper-parameter. We vary it, and use the probability density function to estimate the color field of DTU scenes given the density field from trained DVGOs. The PSNR results of estimated color fields are reported in Table 5. It shows that a large concentration parameter decreases PSNR. Although $c = 0.01$ is slightly better by 0.01 PSNR, we apply a uniform distribution ($c = 0$) in the main paper for its simplified and more efficient calculation process.

Table 5: Color estimation results on DTU scenes with different concentration parameters.

| Concentration parameter | 0 | 0.01 | 0.1 | 1 | 2 |
|---|---|---|---|---|---|
| PSNR | 29.44 | 29.45 | 29.43 | 28.93 | 27.97 |

### B.1.2  Different SH Degrees

In Table 6, we report the PSNR of estimated color fields using different SH degrees. It shows a clear trend that the PSNR increases with the increase of the SH degree. The computational cost is not obviously affected by the SH degree, but the higher SH degree requires much memory.

Table 6: PSNR of estimated color fields using different SH degrees based on DVGO's density fields.

| SH degree | 0 | 1 | 2 | 3 | 4 |
|---|---|---|---|---|---|
| SH basis | 1 | 4 | 9 | 16 | 25 |
| $256^3$ grid | 26.13 | 26.99 | 27.89 | 28.30 | 28.62 |
| $512^3$ grid | 27.22 | 28.29 | 29.44 | 29.98 | - |

## B.2  More Regularization Results

### B.2.1  Visualization and Comparison of Ray-based Losses

We train the DVGO [15, 27] on a typical scene (DTU scan 40). After training, we calculate common ray-based losses, including the photometric loss, per-point RGB loss [27], background entropy loss

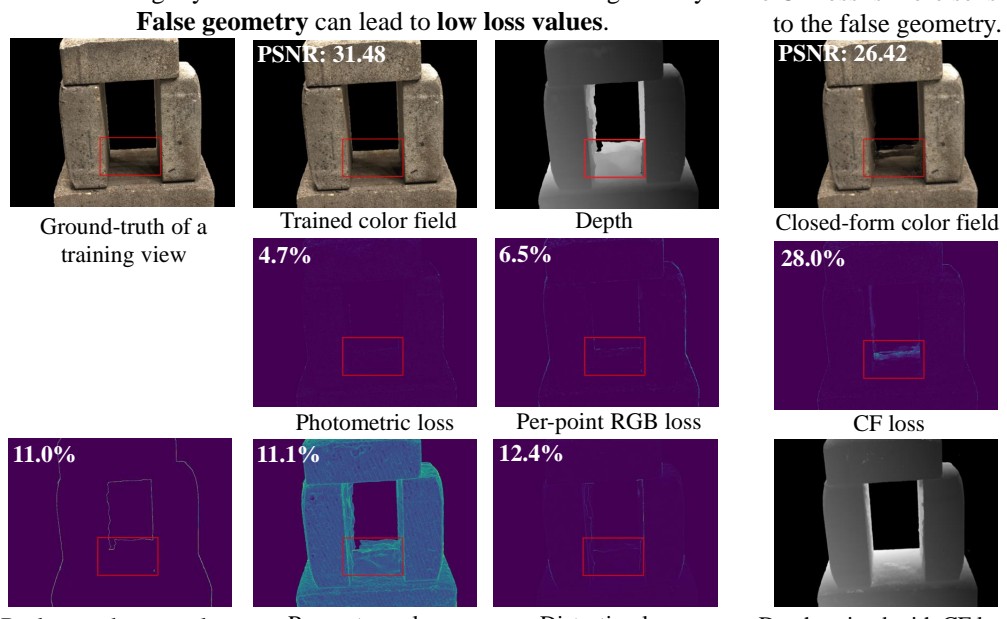

The existing ray-based losses are not aware about the geometry. **False geometry** can lead to **low loss values**. The CF loss is more sensitive to the false geometry.

Ground-truth of a training view    Trained color field    Depth    Closed-form color field

Photometric loss    Per-point RGB loss    CF loss

Background entropy loss    Ray-entropy loss    Distortion loss    Depth trained with CF loss

Figure 12: Visualization and comparison of common ray-based losses on a typical scene with false geometry. All losses are normalized into [0, 1]. The red rectangles mark the false geometry region, and the percentages of the losses in this region over the whole image are shown on the left-top corners.

[15, 27], ray-entropy loss [13], and distortion loss [1], of a training view based on the trained model. For ray-based losses, each pixel has a loss value, so we can visualize the losses as images as shown in Fig. 12. We marked a region with a red rectangle, where the top surface of the bottom rock is falsely reconstructed. On the left-top corners of the loss images, we list the percentages of the losses in the red rectangle over the whole image. These percentages reflect to what extent the corresponding losses focus on the geometric error. The photometric loss of the training view has a very low percentage of 4.7%. This demonstrates that minimizing the photometric loss cannot assure a correct geometry, which is caused by the shape-radiance ambiguity problem. For other ray-based losses, they are also not sensitive enough to the false geometry, with the largest percentage of 12.4%. In contrast, the CF loss has a significantly larger percentage of 28%. It focuses more on geometric errors than common ray-based losses. The errors can be reflected by the image rendered by the closed-form color field, as shown by the right-top image of Fig. 12. The red rectangle has a large closed-form photometric loss, and, more importantly, this loss can be used to improve the geometry of the scene.

### B.2.2    Effect of the Weight Factor $\lambda$ and Number of Rays $|\mathcal{R}_{cf}|$

We vary the two hyper-parameters, including the weight factor and number of rays, on Plenoxels [7], and report the average PSNR and IMRC of the DTU dataset in Table 7. The PSNR and IMRC have an increasing trend when we increase the weight factor. Moreover, we showcase an example scene in Figs. 13-14. By increasing the weight factor or the number of rays, the problem of thick surface has been relieved. When the weight factor is too large ($\lambda = 20$), however, the PSNR decreases. This is because the CF loss dominates the training. While it only provides the supervision signals to the density field, the color field cannot be well optimized. Therefore, the weight factor cannot be too large.

For the number of rays, increasing it helps improve the geometry of the scene. However, more rays mean higher computational cost. To be more specific, 17 minutes are needed on average to train the DTU scenes without CF loss. When we add the CF loss and use 10 rays in every iteration, 32 minutes are needed on average. The time cost further increases to 43 minutes if we use 25 rays. To ensure the training efficiency, we do not increase the number of rays further.

Table 7: Average PSNR↑/IMRC↑ of different weight factors $\lambda$ and numbers of rays $|\mathcal{R}_{cf}|$ on the DTU dataset.

| $|\mathcal{R}_{cf}|$ | $\lambda = 1$ | $\lambda = 2$ | $\lambda = 5$ | $\lambda = 10$ |
|---|---|---|---|---|
| 10 | 32.00 / 16.18 | 32.02 / 16.29 | 32.03 / 16.44 | 32.04 / 16.59 |
| 25 | 32.04 / 16.12 | 32.06 / 16.26 | 32.07 / 16.50 | 32.08 / 16.66 |

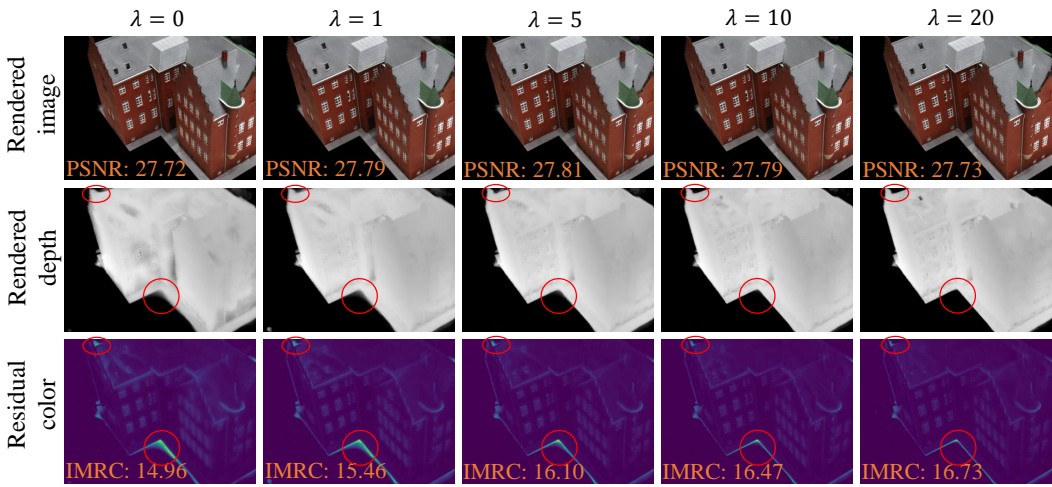

Figure 13: Effect of the weight factor on DTU scan 24.

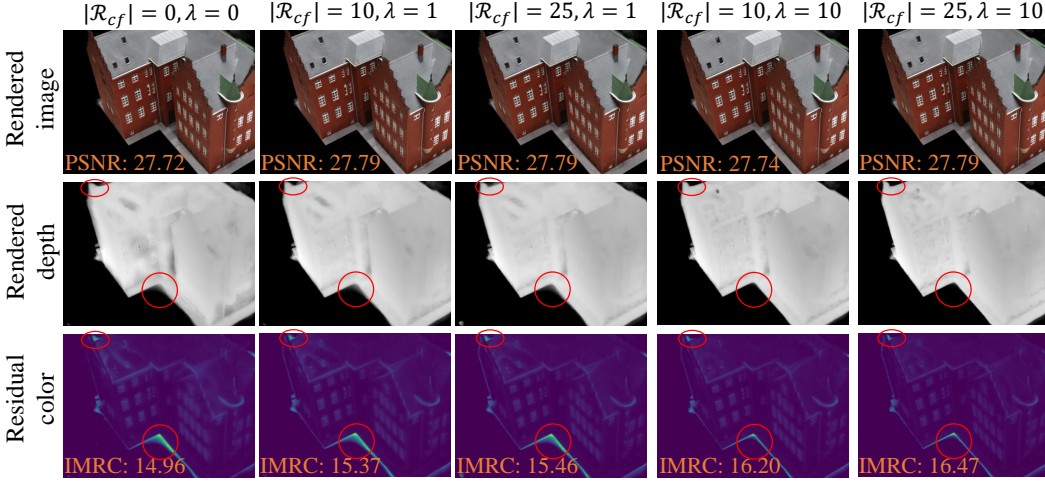

Figure 14: Effect of the number of rays on DTU scan 24.

