# OpenReview forum: "Reducing Shape-Radiance Ambiguity in Radiance Fields with a Closed-Form Color Estimation Method"
_NeurIPS.cc/2023/Conference — NeurIPS 2023 poster_

### Official Review · Reviewer_LJiP · 2023-06-19

**Soundness:** 2 fair
**Presentation:** 3 good
**Contribution:** 3 good
**Rating:** 6
**Confidence:** 5

**Summary:**

This paper use closed-form spherical harmonics (SH) color-field to regularize the training of NeRF-based models. The SH coefficient of each 3D point has a closed-form solution given a set of observed viewing angles. This work further takes the transmittance from the density field to weigh each observed angle. A residual estimation is proposed to reduce the bias due to non-uniform view angles sampling from the training views. The close-form color-field is only used to regularize the optimization of density field. Experiments show that the proposed regularization can improve quality and alleviate shape-raidance ambiguity.

**Strengths:**

Introducing the closed-form SH coefficent solver into NeRF's optimization is an interesting aspect to improve quality of NeRF-based models. I'm interested to see if it is possible to have a closed-form optimal-PSNR color-field in the future given a fixed geometry.

**Weaknesses:**

1. The manuscript incorrectly use the term "volumetric-based" to differentiate voxel-based methods from MLP-based methods (L76). Both of them are volumetric representations mapping spatial coordinates to modality of interest. Please fix it using "implicit" and "explicit" representations instead.

2. Some baseline results are much worse than their official report. The reported PSNR of Plenoxel on NeRF synthetic dataset is 29.83 which is much lower than their official reported 31.71 in Plenoxel's Table2. The baseline DVGO on Mic scene in Fig8 is much worse than their official results where DVGO doesn't produce the floater artifact around the mic wire. I guess the reason is the background color (from solid white to solid black) which make the shape-radiance ambiguity more severe on some scenes.
It is neccessary to have a discussion about the discrepancy of the implementations and the results comparing to the original baselines.

3. The proposed closed-form color-field solution is point-wise without considering the colors from the other points on a ray. As a result, the solved color-field may not be the optimal regarding the photometric loss. This may be one of the reason why Plenoxel (which uses SH as well) have a worse PSNR when using the proposed closed-form solution.

4. Lack of theoratical backup about the bias reduction using residual. The eliminated components (supp's Eq.6) itself is estimated (supp's Eq.10) and is biased due to the non-uniform sampling. It is not clear to me how the overall bias is reduced by using the residual.

5. The ablation study only use a single scene (Fig.4). More examples can make it more convincing.

6. The quantitative improvement is incremental while it needs much more training time (1 hour vs. 10 minutes) and specialized CUDA implementation.

**Questions:**

1. I still don't quite convinced why Plenoxel's color field get much worse PSNR when using the proposed closed-form solver comparing to the SGD solver (Fig5,L235-240). As the original Plenoxel use SH as well, I expect the closed-form solver to have a even better or similar MSE (PSNR) under the same fixed density field. Are there any other factors like the degree of SH or the grid resolution affecting the results? More discussions are necessary.

2. Missing experiments for the number of SH degree. The only discussion is Fig9 showing failure cases on high-frequency view-dependent effect. What is the quality improvement and computation cost of using more or less SH degree?

3. This work assumes the view sampling is uniform so the Monte Carlo integrator (Eq.5) has large error when the training viewing angles is not uniform (Fig.3). However, we actually know the camera poses. Is it possible to derive a pdf over viewing angles using the camera poses so that we can have a correct Monte Carlo integrator like: $$\frac{1}{K} \sum_{k=1}^K \frac{f(x_k)}{\mathrm{pdf}(x_k)} ~.$$

**Limitations:**

The weakness doesn't discussed the discrepency between the point-wise assumption and the volume rendering (see weakness 3).

If the authors decide to give up theoratical proof for the proposed bias reduction strategy, it should also be listed as one of the limitation.

---

> ### Author Rebuttal · Authors · 2023-08-09
>
> # Weaknesses
>
> __W1__: Thank you very much to point out that. We will carefully examine and fix these incorrect terms.
>
> __W2__: Thanks for your comments. The goal of this work is to study the shape-radiance ambiguity problem of the NeRF model. The black background is a more challenging setting, with which more floaters tend to generate in the free space and in some scenes the shape radiance ambiguity problem becoms more severe. That is why the PSNR values drop as you mentioned.
>
> We would like to emphasize that all experiments are done with the same black background images. Thus, the comparison of our method with the baselines is fair under the same challenging setting. We will explain this in the final version of the paper.
>
> __W3__: Thanks a lot for your thoughtful comments. It is true that the closed-form color-field solution is point-wise. Our assumption is that, if a point is not occluded, or it is on the surface of an object, its color along a direction is directly determined by the corresponding observation color in the image. This assumption is true for the ideal case. It is mentioned in the NeRF++ paper that: ideally, the density should peak at the ground-truth surface location, in which case the color reduces to the surface light field. By using this assumption, the proposed closed-form color-field solution becomes sensitive to false geometry. If the geometry or the density field is not correct or sharp enough, which breaks this assumption, the color estimation results will be worse. This leads to worse rendering colors and so a higher CF loss. We think that it is an advantage instead of weakness, because the worse PSNR or high CF loss in the training will correct the false geometry and lead it to more ideal one through backpropagation.
>
> __W4__: We apologize that we cannot provide a rigorous proof this time. We will try to derive the proof in the future and add this to limitations this time.
>
> __W5__: Thanks for your suggestion. We conduct the ablation study on all scenes of DTU given the density volumes from the trained Plenoxels. Below, we can see that by using the residual color estimation, the PSNR improves considerably. By only add the occlusion handling, the PSNR decreases, but when it is combined with the residual color estimation, the PSNR is further improved. This demonstrates the effectiveness of both occlusion handling and residual color estimation.
>
> __Ablation study results__
> |Method|PSNR|
> |:-:|:-:|
> |w/o occlusion handling and residual color estimation|14.61|
> |w/ occlusion and w/o residual color estimation|13.57|
> |w/o occlusion and w/ residual color estimation|25.99|
> |w/ occlusion and residual color estimation|26.49|
>
>
> __W6__: Thanks for your comments. We additionally calculate the PSNR on depth below for the NeRF synthetic data as it provides ground truth depth. It shows that we achieve appreciable improvement.
>
> __Comparison of the PSNR on depth of the NeRF synthetic dataset__
> |Method|Plenoxels|Plenoxels + CF loss|DVGO|DVGO + Distortion loss|DVGO + CF loss|DVGO + Distortion loss + CF loss|
> |:-:|:-:|:-:|:-:|:-:|:-:|:-:|
> |PSNR on depth|22.70|23.08|25.46|25.68|26.36|26.68|
>
> We report the detailed computation cost below.
>
>  __Mean computational cost using different number of CF rays__
> |Number of CF rays|DTU|NeRF synthetic|LLFF|
> |:-:|:-:|:-:|:-:|
> |0|17 min 43 sec|15 min 59 sec|24 min 11 sec|
> |10|32 min 39 sec|20 min 29 sec|29 min 22 sec|
> |25|43 min 1 sec|25 min 32 sec|34 min 33 sec|
>
> # Questions
>
> __Q1__: Thanks for your questions. As we explain in the weakness 3 __W3__, the closed-form solver performs worse because it is more sensitive to false geometry due to the point-wise estimation.
>
> __Q2__: We conduct both color estimation and training experiments about SH degrees and report results below. The increase of SH degree does produce better estimation and training results. The computation cost does not vary much, because the SH bands are efficiently handled by CUDA warps.
>
> __Traininng reuslts with different degrees of SH coefficients on the DTU dataset__
> |SH degree / SH basis|0 / 1|1 / 4|2 / 9|3 / 16|
> |:-:|:-:|:-:|:-:|:-:|
> |PSNR|31.79|31.95|32.08|32.12|
> |IMRC|16.43|16.56|16.66|16.73|
> |Training time|46 min 31 sec|44 min 42 sec| 43 min 1 sec|44 min 36 sec|
>
> __Color estimation results with different degrees of SH coefficients on the DTU dataset__
> |SH degree|0|1|2|3|
> |:-:|:-:|:-:|:-:|:-:|
> |PSNR|27.22|28.29|29.44|29.98|
>
>
> __Q3__: Thanks a lot for your constructive question. A common choice for modeling the distribution of 3D directions is to use the von Mises-Fisher (vMF) distribution defined as follows,
> \begin{equation}
> v(\mathbf{d}; \mu, c) = \frac{c}{2\pi(e^c - e^{-c})} e^{c \mu^\top \mathbf{d}},
> \end{equation}
> where $\mu$ is the nomarlized mean direction, and $c$ is the concentration paramter. Because the vMF distribution only deals with the unimodal case. We resort to the mixture of von Mises-Fisher distribution defined as
> \begin{equation}
> p(\mathbf{d};\mathbf{d}_1, ..., \mathbf{d}_K, c) = \frac{1}{K} \sum_k  v(\mathbf{d}; \mathbf{d}_k, c),
> \end{equation}
> where $\mathbf{d}_1, ..., \mathbf{d}_K$ are known viewing directions as the modes of the distribution. It is noteworthy that when $c$ approches zero, the PDF becomes a uniform distribution. So actually the method used in the paper is a special case of the mixture of vMF  distribution. We vary $c$, and use this PDF to estimate the color field given the density field from trained DVGOs. The results are reported below. The PSNR does not increase much, possibly because the viewing directions are few in our study. Despite this, we believe that it is an important improvement and will further study other PDFs and datasets in the future.
>
> __Color estimation results with a modified Monte Carlo integrator on the DTU dataset__
> |Concentration parameter|0|0.01|0.1|1|2|
> |:-:|:-:|:-:|:-:|:-:|:-:|
> |PSNR|29.44|29.45|29.43|28.93|27.97|

---

> > ### Comment · Reviewer_LJiP · 2023-08-19
> >
> > **W2:** As I mentioned in the discussion in Reviewer qn5t feedback, I think that the claiming about "black background is a more challenging setting" is premature. But the response have addressed my main concern about the fairness of the comparison.
> >
> > **W6:** Thanks for the new results. The improvements are still incremental to me with the extra training time and implementation complexity. However, I find the idea interesting with future potential so it doesn't affect my rating.
> >
> > **Q3:** Thanks for the insightful discussion. I believe that including this discussion in the main paper would be beneficial.
> >
> > I appreciate the author's responses which address my main concerns so I increase my rating. I don't have further questions.

---

### Official Review · Reviewer_Siz9 · 2023-06-22

**Soundness:** 3 good
**Presentation:** 3 good
**Contribution:** 3 good
**Rating:** 6
**Confidence:** 5

**Summary:**

While NeRFs achieve photorealistic results, they suffer from the shape and radiance ambiguity that is inherent in joint reconstruction. To alleviate this issue, the authors aim to separate the estimations using a closed-form radiance estimation. They leverage Spherical Harmonics to represent the radiance given a density field. With explicit occlusion handling based on the density field, all cameras with visibility for the point can be detected. The colors are then used to fit the SH coefficients for the specific surface point. The authors propose to use this closed-form color to regularize the NeRF training by enforcing similar behavior between the NeRF output and the closed-form color besides the regular photometric loss.

The effectiveness of this formulation is shown in various experiments and is agnostic to the underlying NeRF method used.

**Strengths:**

- One of the most significant advantages of this method is that the regularization is universally applicable to any NeRF method.
- The method introduces a nice way to add a soft global view-dependent color regularization.
- Besides issues with highly reflective surfaces, it improves considerably.

**Weaknesses:**

- Currently, I have the feeling the paper would benefit from an overview figure. This can also be beneficial as a reference in the experiments, where some evaluations only rely on the closed-form color, but in others, it's used as a regularization. Here, one can create a direct link to the overview figure and the symbols for each color output. A simple overview of how the method slots in an existing NeRF framework would be a great addition and can be easily incorporated next to Fig. 1.
- The authors compared against other ray distribution priors but did not compare with model priors such as RegNeRFs depth/normal input from omnidata.
- Similarly, RefNeRF also aims to improve surface reconstruction. How does this method fare against the proposed one?
- The authors should briefly discuss Riegler et al. - Stable View Synthesis, as it also handles aggregation of features from multiple views. The approach is vastly different, but the rough conceptual idea remains.
The authors mentioned that the closed-form color field has issues approximating highly reflective objects, but the limitations are also quite severe. The performance of anything which drastically derivates from Lambertian reflection will suffer. Biasing everything towards Lambertian also solves the shape/radiance ambiguity. So I would like to see if experiments with only lower frequency view encodings or a RefNeRF with an l2 penalty on the estimated roughness are used.

Minor:
- In 81/82: NeRF’s representation _of_ a scene
- In 82: remove ‘makes it’: The NeRF’s representation of a scene inherently suffers from shape-radiance ambiguity
- In 83: error _in_ geometry.
- In 86: needs to be solved
- In 174, the authors used minus as a verb. The correct verb is: subtract

**Questions:**

- What is the additional overhead of the closed-form color calculation?
- Would the decrease in performance on highly reflective surfaces be mitigated with more SH bands, or would the method become instable during training?

**Limitations:**

See the last point in the weakness section.

---

> ### Author Rebuttal · Authors · 2023-08-09
>
> # Weaknesses
>
> __W1__: Thanks a lot for your useful suggestion. We will carefully prepare an overview figure that clarifies which part of the experiments is about evaluation and which is about regularization. We will also add links to the symbols, and how the method slots in existing NeRFs.
>
> __W2__ & __W3__: Thanks a lot for you comments. To better compare with RegNeRF and RefNeRF, we additionally train the vanilla NeRF model and report the results below. The hyper-parameters are the same as the deafault values in original implementation, and all the input data are the same as the one we use, which makes it a fair comparison. For the implicit model, the RegNeRF and RefNeRF outperform NeRF due to their regularization and special formulation. Their PSNRs also outperforms those of Plenoxels and DVGO. But they need much longer training time. Specifically, more than one day is needed for training a scene using RegNeRF and RefNeRF, while the explicit models requires only less than an hour. Furthermore, the explicit models also enable faster rendering in the testing phase.
>
> __More experiments on other models on the DTU dataset__
> |Model|PSNR|IMRC|Training time|
> |:-:|:-:|:-:|:-:|
> |NeRF|31.95|17.95|>10 hours|
> |RegNeRF|32.41|18.90|>1 day|
> |RefNeRF|32.34|18.61|>1 day|
>
> __W4__: Thank you very much to provide us the omitted related work. The Stable View Synthesis encodes features of images from source views by using a convolutional network and aggregates the features from these views for predicting the target view. The conceptual idea is roughly the same. We will add further discussion about it to the related work section.
>
> For the lower frequency view encodings and the issues about the highly reflective objects. Please refer to the answer of __Q2__ below for a detailed discussion.
>
> # Questions
>
> __Q1__: Thanks for your comments. We would like to breifly introduce our implementation first. At each batch of training, a batch of $B$ rays are used. Then, only the SH coefficients of the voxels intersected with these rays need to be estimated. Suppose that each ray involves of $K$ voxels ($K$ varies for different rays actually), there are $M=BK$ voxels. To estimate the SH coefficients of a voxel, the density values along the ray that connects the voxel to each source cameras need to be calculated. Suppose that there are $N$ cameras, then the total number of voxels involved is $NMK=NBK^2$. In our implementation, each CUDA block handles one of the $M$ voxels, and each thread in the block handles one of the $N$ cameras. So each thread handles a ray from a voxel toward each source camera, which involves the calculation of the order of $K$ density values. As the threads are executed in a parallel way, in theory, this reduces the computation overhead from the order of $NBK^2$ to $K$. In practice, we observe that sometimes the computational overhead is affected by the batch size $B$. This is due to the hardware limit of the maximum CUDA threads. When too much threads are used, they are not guaranteed to be executed in a parallel way. We report and compare the computational cost of our method below.
>
> __Mean computational cost using different number of CF rays__
> |Number of CF rays|DTU|NeRF synthetic|LLFF|
> |:-:|:-:|:-:|:-:|
> |0|17 min 43 sec|15 min 59 sec|24 min 11 sec|
> |10|32 min 39 sec|20 min 29 sec|29 min 22 sec|
> |25|43 min 1 sec|25 min 32 sec|34 min 33 sec|
>
> The number of source cameras used are the number of training images. They are 44 or 58 for DTU, 100 for NeRF synthetic, and from 17 to 54 for LLFF. The DTU requires longer training time because it is trained for 12 epochs, while the NeRF synthetic is trained for 9 epochs. For 9 epochs, the DTU cost about 24 minutes and 32 minutes for 10 and 25 rays, respectively. Compare these with those of NeRF synthetic, we can observe that the large number of 100 cameras does not obviously increase the computational burden.
>
> __Q2__: First of all, we would like to clarify that there are two kinds of results listed in the paper. One is the color estimation results based on the density volume from pre-trained models. The other is the training results of the models armed with the CF loss. We apologize for the confusion and will decribe this more clearly in the final version. The results in Figure 9 is the color estimation results given the density volume from a pre-trained model. It does not involve a training process. In case of the training results, we would like to emphasize that the PSNR for the scene in Figure 9 is improved by using the CF loss. In case of the color estimation process, the PSNR of a closed-form color field does increase by using more SH bands as you have mentioned. Below we report the average PSNR over all DTU scenes from SH degree 0 to 3 (We use degree 2 in the paper).
>
> __Color estimation results with different degrees of SH coefficients on the DTU dataset__
> |SH degree|0|1|2|3|
> |:-:|:-:|:-:|:-:|:-:|
> |PSNR|27.22|28.29|29.44|29.98|
>
> We also train with lower and higher frequency view encodings, which aim to answers the weakness 4 __W4__ you point out. The lower view encodings deal with non-Lambertian effects worse, and so the PSNR is worse. By training with more SH bands with degee 4, the results become better.
>
> __Traininng reuslts with different degrees of SH coefficients on the DTU dataset__
> |SH degree / SH basis|0 / 1|1 / 4|2 / 9|3 / 16|
> |:-:|:-:|:-:|:-:|:-:|
> |PSNR|31.79|31.95|32.08|32.12|
> |IMRC|16.43|16.56|16.66|16.73|
> |Training time|46 min 31 sec|44 min 42 sec| 43 min 1 sec|44 min 36 sec|

---

> > ### Comment · Reviewer_Siz9 · 2023-08-21
> >
> > Thank you for the detailed rebuttal. I have no further questions and will increase my rating.

---

### Official Review · Reviewer_hBrk · 2023-07-07

**Soundness:** 2 fair
**Presentation:** 1 poor
**Contribution:** 2 fair
**Rating:** 4
**Confidence:** 4

**Summary:**

While nerfs have shown impressive properties in the ability to reconstruct the geometry of three dimensional scenes, there are still some cases where the performances are not as good as expected. This paper focuses on one of these problems, the untangling of the geometry from the color. For that, the authors propose to learn a second color model, using a close form formula based on a spherical harmonics model. On top of the classic ray tracing of nerf, this method requires to trace additional rays that goes from the query point to the cameras. The authors also show the existence of a bias depending on the distribution of view angle and propose a method to correct it.

**Strengths:**

The paper focus on an important problem. Estimating accurately the geometry in complex scene, especially with high specularity, is still a very relevant today.

Despite the heavy computational load, the authors were able to successfully train different models. They also provide results on three different datasets, with two of them being real images.

**Weaknesses:**

The premise of the method is that density and colors are estimated independently in a nerf model. This is actually false. In particular, in the vanilla nerf model the prediction of the color directly depends on the density. The color MLP takes as input not the coordinates of the point but the features corresponding to the density predicting network. This means that nerf should actually be modeled by (c) in Figure 1 and not (a) as claimed in the paper.

Many variables are not introduced properly making the reading more difficult that it should.

The method is highly inefficient. For a given ray, for a given sampled point, the method requires to integrate the density along the ray toward each source camera. This means that if the model is trained using $N$ cameras and $M$ points are sampled each time to approximate the integrals, the proposed method requires to estimate the density of $NM^2$ points instead of the usual $M$ points.

The parameters of the method vary between datasets. No explanation or intuition is provided for that.

Regarding the experiments, the results are quite incremental with an increase of the order of 0.1dB compared to two previous methods. It's also disappointing that the analysis was not performed with more models. We can especially mention tensoRF that also has a model based on SH. It would have also been interesting to compare to methods like regnerf that adds additional regularizers to nerf to be able reconstruct better the geometry even in difficult conditions. Moreover, while the paper focuses on improving the geometry of the trained model, there is no quantitative experiment showing that the proposed method has indeed an impact on the geometry thus learned. Only a few qualitative results are presented with a color map for the depth that is very difficult to interpret. One can also regret the lack of computational analysis. No study are done on the unbiaising process proposed in the paper.

**Questions:**

What strategy of point sampling is performed for estimating the color and transmittance of a given point (i.e when the ray from this point to the camera is drawn)?

**Limitations:**

I don't foresee any potential negative societal impact of the work.

---

> ### Author Rebuttal · Authors · 2023-08-09
>
> # Weaknesses
> __W1__: Thanks for your comments. The NeRF can be categorized into implicit and explicit models. We focus on the explicit ones in this work. They model the density and color as volumes. The color volume does not receive features from the density one, and so the they can be estimated independently. We believe that the premise of our method is correct.
>
> __W2__: Thanks for your feedback. We will carefully revise the variables.
>
> __W3__: Thanks for your comments. To relieve the computational burden, we implement by CUDA kernels. At each batch of training, a batch of $B$ rays are used. Then, only the colors of the voxels intersected with these rays need to be estimated. Suppose that each ray involves $K$ voxels ($K$ varies for different rays actually), there are $M=BK$ voxels. To estimate the color of a voxel, the density values along the ray from the voxel to $N$ source cameras need to be calculated. Then the total number of voxels involved is of order $NMK=NBK^2$. In our implementation, each CUDA block handles one of the $M$ voxels, and each thread handles one of the $N$ cameras. As the threads are executed in a parallel way, in theory, this reduces the computation overhead from the order of $NBK^2$ to $K$. In practice, we observe that sometimes the computational overhead is affected by the batch size $B$. This is due to the hardware limit of the maximum CUDA threads. When excessive threads are used, they are not guaranteed to be executed parallelly. We report and compare the computational cost below. The computational overhead is acceptable.
>
> __Mean computational cost using different number of CF rays__
> |Number of CF rays|DTU|NeRF synthetic|LLFF|
> |:-:|:-:|:-:|:-:|
> |0|17 min 43 sec|15 min 59 sec|24 min 11 sec|
> |10|32 min 39 sec|20 min 29 sec|29 min 22 sec|
> |25|43 min 1 sec|25 min 32 sec|34 min 33 sec|
>
> The number of source cameras used are 44 or 58 for DTU, 100 for NeRF synthetic, and from 17 to 54 for LLFF. The DTU requires longer training time because it is trained for 12 epochs, while the NeRF synthetic is trained for 9 epochs. For 9 epochs, the DTU cost about 24 minutes and 32 minutes for 10 and 25 rays, respectively. Compare these with those of NeRF synthetic, we can observe that the large number of 100 cameras does not obviously increase the computational burden.
>
> __W4__: Thanks for your comments. To determine the hyperparameters, we set aside a portion of training data as validation data, and use grid search to find the best hyperperameters. Then, we use all training data for training. Specifically, we try the weight factor for the CF loss including 1, 5, 10, and 20, and the number of rays including 10 and 25.
>
> __W5__: Thanks a lot for your detailed comments. We separate our answer to four sections in the following.
>
> __W5.1 More expriments__
>
> We follow the default hyperparamter setting in the original work, and train TensoRF, TensoRF with CF loss, NeRF, RegNeRF, and RefNeRF on the DTU dataset. The results are reported below. For the implicit model, the RegNeRF and RefNeRF outperform NeRF due to their regularization and special formulation. Their PSNRs also outperforms those of Plenoxels and DVGO. But they need much longer training time. The explicit model TensoRF performs better, because it allows a high grid resolution due to its decomposition of density and color volumes. By adding CF loss to the TensoRF, the PSNR further improves. This demonstrates the effectiveness of our method.
>
> __More experiments on other models on the DTU dataset__
> |Model|PSNR|IMRC|Training time|
> |:-:|:-:|:-:|:-:|
> |NeRF|31.95|17.95|>10 hours|
> |RegNeRF|32.41|18.90|>1 day|
> |RefNeRF|32.34|18.61|>1 day|
> |TensoRF|32.49|18.86|22 min 56 sec|
> |TesorRF + CF loss|32.66|19.04|57 min 23 sec|
>
> __W5.2 Quantitative results of geometry evaluation__
>
> We evaluate the geometry of the models trained on the NeRF synthetic dataset by calculating the PSNR on depth maps. The results are reported below. Our CF loss increases the PSNR by 0.38 for the Plenoxels and 0.9 for the DVGO. For the datasets that do not have ground truth depth maps, we use the IMRC metric defined in equation (14) in the manuscript as an alternative. As Reviewer qn5t points out, this metric is an approximation of PSNR in 3D. The wrong geometry, such as floaters in the free space, will be penalized due to the high residual color. Our method increases this metric on both DTU and LLFF dataset. Overall, these quantitative results demonstrate that it makes geometry better.
>
> __Comparison of the PSNR on depth of the NeRF synthetic dataset__
> |Method|Plenoxels|Plenoxels + CF loss|DVGO|DVGO + Distortion loss|DVGO + CF loss|DVGO + Distortion loss + CF loss|
> |:-:|:-:|:-:|:-:|:-:|:-:|:-:|
> |PSNR on depth|22.70|23.08|25.46|25.68|26.36|26.68|
>
> __W5.3 Computational analysis__
>
> Please see the answer to weakness (__W3__) you point out.
>
> __W5.4 Study about the unbiasing process__
>
> We conduct ablation studies for the color estimation process given the density from trained Plenoxels, which involves two components including occlusion handling and residual color estimation or unbiasing. The results are reported below. Please refer to the answer to the weakness 5 (W5) pointed out by the Reviewer LJiP for a detailed discussion. This study demonstrate the effectiveness of the unbiasing process.
>
> __Ablation study of the closed-form color estimation on the DTU dataset__
> |Method|PSNR|
> |:-:|:-:|
> |w/o occlusion handling and residual color estimation|14.61|
> |w/ occlusion and w/o residual color estimation|13.57|
> |w/o occlusion and w/ residual color estimation|25.99|
> |w/ occlusion and residual color estimation|26.49|
>
>
> # Questions
>
> __Q1__: Thanks for your question. The point sampling strategy for the rays that connect this point to a camera center is the same as the one used in original training rays in Plenoxels. Specifically, it uses uniform sampling along the ray that intersects with the density volume with a step size of 0.5 voxel, and no random noise is applied in the sampling.

---

### Official Review · Reviewer_qn5t · 2023-07-14

**Soundness:** 2 fair
**Presentation:** 3 good
**Contribution:** 3 good
**Rating:** 7
**Confidence:** 4

**Summary:**

The paper proposes a novel regularization technique for optimizing radiance fields to achieve more accurate geometry alongside realistic novel views. The regularizer is based on a closed-form estimation of spherical harmonic coefficients for view-dependent color, as a function of the training pixel colors and the predicted density field. Using this closed-form procedure to estimate color from density enables regularizing density directly through photometric loss. This regularizer is therefore adaptive to each scene, unlike more general, standard scene-agnostic regularizers like sparsity and total variation. The paper shows empirical improvement in reconstructed geometry via improved rendered depth maps, for two voxel-based scene representations (DVGO and Plenoxels).

**Strengths:**

I really like the observation the paper makes that it is possible to predict color in closed form as a function of density field and training ray colors. It makes a lot of sense that using this closed-form solution for color and then optimizing density directly through photometric loss, should yield more accurate reconstructed density fields. And indeed the results do show improved depth renderings.

**Weaknesses:**

presentation weaknesses
- Figure 2 should include o_k and P_k, and ideally also F_c. In its current form there is a bit of a struggle to match variables in the text to components in the figure.
- I don’t see much value added by Figure 3. It is used to illustrate the idea that nonuniform or systematically biased sampling can result in a systematic error in the estimated integral. In my view, this idea is clear from textual description, and the space currently dedicated to the figure could be better spent on detail (either textual or ideally graphical) about how the proposed bias correction works.
- As an added point of confusion around Figure 3, why include a term that is cos(0x) (in both the figure and the accompanying text)? It appears that this is literally referring to cos(0)=1, which has no effect in the example shown.
- Equation 14 (the definition of the IMRC metric) could benefit from another sentence of explanation, describing the intuition that this metric is an approximation of PSNR in 3D.
- Figures 4 and 5 would benefit from more substantial captions that explain the experiment being presented and how the results should be interpreted. Currently this information is only available in the text, but it’s best if figures are also self-contained.
- It’s not clear from the paper whether the results (or which subset of the results) are fine-tuned using pre-trained models vs trained from scratch. I’m not sure if the argument is that fine-tuning with the closed-form loss improves existing pre-trained geometry, or that training from scratch with the closed-form loss as a regularizer produces better geometry, or both.
- There are scattered typos and minor grammatical issues; please copy-edit the final paper.
evaluation weaknesses
- The evaluation uses a black background, whereas in the baseline papers a white background was used. Is there a reason for changing the background color?
- PSNR values reported for the two baselines (Plenoxels and DVGO) are substantially lower than the values reported in the original papers (29.83 vs 31.71 for Plenoxels, and 31.58 vs 31.95 and 32.8 for DVGO). As far as I’m aware the only difference is the use of the black background; I’m concerned that either this or some other implementation difference makes the baselines used unnecessarily weak.
- I understand the IMRC metric and see why it’s useful for datasets where ground truth geometry is unavailable, but in my view an even stronger evaluation would be to compare PSNR on depth maps for a dataset where ground truth geometry is available (e.g. this could be rendered for the NeRF-synthetic dataset).
- Figure 9 shows that there is a gap in quality between renderings produced by the trained color field vs the closed-form color field—can the authors comment on the origin of this gap, and how it might be closed?

**Questions:**

My biggest question about this paper is why the proposed method is used as a regularization scheme on top of the usual least squares objective, rather than optimizing a density-only model and directly using the colors from the closed-form predictions. I wouldn’t be surprised if doing so yields even better geometry, and reduces model size as an added bonus. Or if this simpler strategy works worse than what is proposed, do the authors have an explanation why (perhaps related to the limitation illustrated in Figure 9)?

I would also appreciate responses to my comments in the weaknesses section, primarily regarding evaluation weaknesses.

**Limitations:**

Yes, limitations are discussed to some extent, but the paper would benefit from more discussion of the origin (and potential remedies, if they exist) of the limited quality of the estimated color field.

---

> ### Author Rebuttal · Authors · 2023-08-09
>
> #  Presentation weaknesses
>
> Thank you very much for your detailed comments and suggestions about the presentation. We will try our best to fix these in the final version of the paper.
>
> # Evaluation weaknesses
>
> __W1__: Thanks for your comments. The goal of this work is to study the shape-radiance ambiguity problem of the NeRF model. The black background is a more challenging setting, with which more floaters tend to generate in the free space. It provides a better way to reveal and study the shape-radiance ambiguity problem. To make it clearer, we will add this reason in the final version of the paper in the section of experimental settings.
>
> __W2__: Thanks for your comments. First, we would like to clarify that there are not any other implementation differences except for the use of the black background. The black background is a more challenging setting, so that the PSNR values drop as you mentioned. We would like to emphasize that all the experiments in the paper are done with the same black background images. Thus, the comparison of our method with the baselines is fair under the same challenging setting. Moreover, in practice, photos may be taken at night, which makes their background mainly black, so it is also meaningful to study the model performance under this setting. To make readers clearer, we will explain this in the final version of the paper in the section of experimental results.
>
> __W3__: Thank you very much for providing us a stronger evaluation method. We evaluate the depth maps for the NeRF-synthetic dataset, as the ground truth depth is provided in the test datasets. Because the pixel values of an image for PSNR calculation fall within [0, 1], we first normalize the ground truth and predicted depth to be within this range. Then, we can calculate the PSNR on depth maps. The results are reported in Table below. We can see that for both Plenoxels and DVGO, the CF loss improves the quality of the depth maps.
>
>  __Comparison of the PSNR on depth of the NeRF synthetic dataset__
> |Method|Plenoxels|Plenoxels + CF loss|DVGO|DVGO + Distortion loss|DVGO + CF loss|DVGO + Distortion loss + CF loss|
> |:-:|:-:|:-:|:-:|:-:|:-:|:-:|
> |PSNR on depth|22.70|23.08|25.46|25.68|26.36|26.68|
>
> __W4__: It is difficult to model highly reflective object with low SH degree, especially given few views. The drop in PSNR can be greatly mitigated by using a higher SH degree. Below we report the average PSNR over all DTU scenes from SH degree 0 to 3 (We use degree 2 in the paper). In average, the PSNR increase by 0.54 dB from SH degree 2 to 3.
>
> __Color estimation results with different degrees of SH coefficients on the DTU dataset__
> |SH degree|0|1|2|3|
> |:-:|:-:|:-:|:-:|:-:|
> |PSNR|27.22|28.29|29.44|29.98|
>
>
> # Questions
>
> __Q1__: Thanks a lot for your constructive question. We are also aware that it is possible to optimize a density only model directly using our proposed CF loss. The main difficulty lies on the computational overhead. We give a computational analysis in the answer to the Weakness 3 to the Reviewer hBrk, and report the computational cost in the Table below. A main conclusion is that the computational cost is affected by the batch size or number of rays in practice. In our experiments, we use 25 rays in each batch for regularization in addition to 5000 rays used in the original optimization process, which takes about 43 minutes to train a DTU model in average.
>
> __Mean computational cost using different number of CF rays__
> |Number of CF rays|DTU|NeRF synthetic|LLFF|
> |:-:|:-:|:-:|:-:|
> |0|17 min 43 sec|15 min 59 sec|24 min 11 sec|
> |10|32 min 39 sec|20 min 29 sec|29 min 22 sec|
> |25|43 min 1 sec|25 min 32 sec|34 min 33 sec|
>
> On one hand, if we increase the number of rays to 5000 for either regularization or optimization for a density only model. The computational cost will be increased from below an hour to a half day. As we focus on the explicit NeRF models in this work, which feature a fast training process, we aim to limit the training time to be within an hour. On the other hand, if we only use 25 rays in each batch to optimize a density only model, the results are worse than the original model that uses 5000 rays in each batch. That is why we apply the CF loss as a regularization.
>
> It is still interesting to compare the density only model with the original model under a fair setting. Specifically, we train the original Plenoxels also using 25 rays on the DTU dataset, and compare it with the density only model trained by the CF loss also using 25 rays. This is a more challenging setting, as few training rays easily make the model overfit to a local region, and thus exacerbate the shape-radiance ambiguity problem. We report the mean metrics in the Table below. The density only models not only converge, but even yield better PSNR and IMRC.
>
> __Comparison of the Plenoxels and density only model with CF loss by using 25 rays__
> |Method|PSNR|IMRC|
> |:-:|:-:|:-:|
> |Plenoxels|26.12|14.21|
> |Density only model with CF loss|27.82|15.33|
>
> In summary, we believe that there is great potential to train density only models using the CF loss, but in this work, we use it as a regularization term to meet our goal of acceptable training time. The unique superiority of our regularization method is that even with only the regularization term, the model can converge, which is not achieved by other regularization in NeRF to the best of our knowledge. We will further explore how more CF rays work in the future.

---

> > ### Comment · Reviewer_qn5t · 2023-08-13
> >
> > Thanks to the authors for the detailed response and additional experiments.
> >
> > I greatly appreciate the comparison of training with and without closed-form color only, rather than using it as a regularizer. Even though this comparison is done using very small batches for computational reasons, it still shows promising results. I would very much like to see a strategy that optimizes density only and retains computational efficiency, but it’s ok if this is deferred to future work.
> >
> > I still don’t really "buy" the explanation that black background is a more challenging setting than white background; my suspicion is that it is only more challenging because the prior methods use parameters that were tuned to work well with a white background, and re-tuning for a black background may well recover similar performance. Or at least this has been my experience in some of my own experiments.

---

> > > ### Author Response · Authors · 2023-08-17
> > >
> > > Thanks very much for your reply and thoughtful comments! We fully understand your concern about the parameters. Therefore, we carefully search for the weight factors of total variation loss for both density ($tv_d$) and color volumes ($tv_c$) on the NeRF synthetic dataset. They are the most important hyper-parameters. Other hyper-parameters such as the learning rate, grid size, and number of training epochs remain the same. Specifically, the $tv_d$ ranges from 1e-5 to 1e-2, and the $tv_c$ ranges from 1e-4 to 1e-1. Below, we report the results that are centered around the best one (highlighted in bold face).
> > >
> > > __Grid searched PSNRs for different combinations of $tv_d$ and $tv_c$__
> > > |$tv_d$ \ $tv_c$| 1e-4 | 1e-3 | 1e-2 | 1e-1|
> > > |-----------|-----|-----|-----|-----|
> > > |__1e-5__|29.70|29.83|29.71|29.14|
> > > |__1e-4__|29.80|29.92|29.88|29.33|
> > > |__1e-3__|29.70|29.88|__29.97__|29.63|
> > > |__1e-2__|29.31|29.63|29.90|29.63|
> > >
> > > The original results is 29.83 for PSNR and 22.70 for the PSNR on depth. After the grid search, the best PSNR becomes 29.97 and the corresponding PSNR on depth is 22.99. The PSNR does improve, but compared with the PSNR trained on the white background, i.e., 31.71, there is still a large gap. Therefore, we are afraid that re-tuning the hyper-parameters, at least the weight factors of total variation loss, does not recover similar performance.
> > >
> > > Furthermore, we add the proposed CF loss to the best case. The resulting PSNR is 29.99 and the PSNR on depth is 23.62. We can see that under the same setting, the CF loss still helps to improve the geometry of the scenes by 0.63 dB in average. We would like to emphasize again that our comparison is fair, the only difference between our method and the original model is that we add the CF loss. All other settings are the same.
> > >
> > > Thanks again for your reply.

---

> > > > ### Comment · Reviewer_qn5t · 2023-08-18
> > > >
> > > > Thanks for the additional tuning to ensure a fair comparison. I am raising my score in light of (1) I really like the core idea here, which is the recognition that color can be predicted in closed form from density and this can be used to improve geometry reconstruction, and (2) although the paper doesn't go all the way to efficiently optimize a density-only model, it gets a good fraction of the way there and shows enough improvement that I'd like to see the paper accepted and built upon.

---

> > > > ### Comment · Reviewer_LJiP · 2023-08-19
> > > >
> > > > I agree with Reviewer qn5t that the claiming about "black background is a more challenging setting" is premature. I appreciate the new experiments about the tv loss. However, some of the fixed hyperparameters like lr and decay schedule can heavily affect results from my experience. Besides, the reason may be due to the appearance of the few tested objects. Maybe some other objects have better performances with black bg instead of white bg.

---

### Author Rebuttal · Authors · 2023-08-09

Dear all,

We would like to deeply appreciate comments from all reviewers, which are invaluable to improve our paper. We have carefully considered all comments and reponsed in a point-by-point way.

In order to address all concerns and questions raised by reviewers, and make our paper better, we summarize all additional evaluation results based on the reviews and show them as follows. We believe these additional evaluations can help us demonstrate the effectiveness and efficiency of our method.


__Table 1__: Comparison of the Plenoxels and density only model with CF loss by using 25 rays
|Method|PSNR|IMRC|
|:-:|:-:|:-:|
|Plenoxels|26.12|14.21|
|Density only model with CF loss|27.82|15.33|

__Table 2__: Comparison of the PSNR on depth of the NeRF synthetic dataset
|Method|Plenoxels|Plenoxels + CF loss|DVGO|DVGO + Distortion loss|DVGO + CF loss|DVGO + Distortion loss + CF loss|
|:-:|:-:|:-:|:-:|:-:|:-:|:-:|
|PSNR on depth|22.70|23.08|25.46|25.68|26.36|26.68|

__Table 3__: Mean computational cost using different number of CF rays
|Number of CF rays|DTU|NeRF synthetic|LLFF|
|:-:|:-:|:-:|:-:|
|0|17 min 43 sec|15 min 59 sec|24 min 11 sec|
|10|32 min 39 sec|20 min 29 sec|29 min 22 sec|
|25|43 min 1 sec|25 min 32 sec|34 min 33 sec|

__Table 4__: More experiments on other models on the DTU dataset
|Model|PSNR|IMRC|Training time|
|:-:|:-:|:-:|:-:|
|NeRF|31.95|17.95|>10 hours|
|RegNeRF|32.41|18.90|>1 day|
|RefNeRF|32.34|18.61|>1 day|
|TensoRF|32.49|18.86|22 min 56 sec|
|TesorRF + CF loss|32.66|19.04|57 min 23 sec|

__Table 5__: Traininng reuslts with different degrees of SH coefficients on the DTU dataset
|SH degree / SH basis|0 / 1|1 / 4|2 / 9|3 / 16|
|:-:|:-:|:-:|:-:|:-:|
|PSNR|31.79|31.95|32.08|32.12|
|IMRC|16.43|16.56|16.66|16.73|
|Training time|46 min 31 sec|44 min 42 sec| 43 min 1 sec|44 min 36 sec|

__Table 6__: Color estimation results with different degrees of SH coefficients on the DTU dataset
|SH degree|0|1|2|3|
|:-:|:-:|:-:|:-:|:-:|
|PSNR|27.22|28.29|29.44|29.98|

__Table 7__: Color estimation results with a modified Monte Carlo integrator that uses the mixture of von Mises-Fisher distribution on the DTU dataset
|Concentration parameter|0|0.01|0.1|1|2|
|:-:|:-:|:-:|:-:|:-:|:-:|
|PSNR|29.44|29.45|29.43|28.93|27.97|

__Table 8__: Ablation study of the closed-form color estimation on the DTU dataset
|Method|PSNR|
|:-:|:-:|
|w/o occlusion handling and residual color estimation|14.61|
|w/ occlusion and w/o residual color estimation|13.57|
|w/o occlusion and w/ residual color estimation|25.99|
|w/ occlusion and residual color estimation|26.49|

---

### Decision · Program_Chairs · 2023-09-21

**Decision:**

Accept (poster)

**Comment:**

Three reviewers are positive about the paper, in particular, they are excited about the core idea and believe that it will inspire follow-up work. The evaluation missed some baselines and analysis, but the authors' rebuttal provided many of the results. The three reviewers are convinced.

Reviewer hBrk is negative about the paper. The main complaints were the computational overhead, missing experiments with more models like TensoRF, and a slight improvement over the previous methods (given the overhead).

As the reviews do not converge, the AC has to weigh among different opinions/viewpoints. After reading all the comments, the AC believes that the idea of using closed-form estimation of spherical harmonic coefficients for view-dependent color is quite novel and will be beneficial for the community. Its improvement was indeed not significant but is consistent over various commonly used NeRF models. As discussed in several reviewers' comments, such a work will inspire follow-up work in this area. Considering the pros and cons, the AC decided to recommend acceptance.